

# Temporary waterlogging alters CO₂ flux dynamics but not cumulative emissions in cultivated mineral soils

Reija Kronberg[1,2], Sanna Kanerva[1], Markku Koskinen[1,2], Tatu Polvinen[1,2], Tuomas Mattila[3], & Mari Pihlatie[1,2]

5   [1]University of Helsinki, Department of Agricultural Sciences, Unit of Environmental Soil Science, Viikinkaari 9, P.O. Box 56, FI-00014 Helsinki, Finland
[2]Institute for Atmospheric and Earth System Research (INAR)/Faculty of Agriculture and Forestry, University of Helsinki, Viikinkaari 9, P.O. Box 56, FI-00014 Helsinki, Finland
[3]Finnish Environment Institute SYKE, Latokartanonkaari 11, 00790, Helsinki, Finland

*Correspondence to*:

Reija Kronberg

E-mail address: reija.kronberg@helsinki.fi

Postal address: Viikinkaari 9, 00790 Helsinki

Mari Pihlatie

E-mail address: mari.pihlatie@helsinki.fi

Postal address: Viikinkaari 9, 00790 Helsinki



**Abstract**

Increasingly variable rainfall patterns expose soils to more frequent waterlogging in humid climates. Yet, the effects of waterlogging on soil organic matter decomposition in mineral soils remain uncertain. We studied the impact of off-season waterlogging on carbon dioxide ($CO_2$) and methane ($CH_4$) production, and dissolved carbon dynamics in controlled greenhouse conditions using 32 soil profiles (h=63 cm, d=15.2 cm) sampled from two agricultural fields (silty clay, sandy loam) in southern Finland. During the 1.5-year study comprising three growth cycles, spring barley (*Hordeum vulgare*) was grown during the

growing seasons. During the off-seasons, half of the monoliths were subjected to waterlogging episodes of seven weeks, while in the control monoliths soil moisture was maintained below field capacity. Additionally, overwintering cover crop (*Festuca arundinacea*) was grown in half of the monoliths. Soil temperature and moisture were continuously monitored, dissolved organic (DOC) and inorganic carbon (DIC) concentrations in pore water were analyzed at three depths, and $CO_2$ and $CH_4$ fluxes were measured at the surface. Temporary waterlogging did not induce $CH_4$ production in either soil. Contrary to our

hypothesis, waterlogging did not increase soil DOC content. Instead, on-going microbial/rhizospheric activity promoted an increase in DIC content while $CO_2$ fluxes declined, indicating an accumulation of respired $CO_2$ in soil pore water. This sustained $CO_2$ production could not be explained solely by mobilization of Fe-associated C, as initially hypothesized. After the initiation of drainage, $CO_2$ fluxes from both soils and plant treatments increased more than predicted based on changes in soil moisture and temperature, likely due to the release of previously accumulated $CO_2$. These post-waterlogging increases in

$CO_2$ fluxes roughly equaled the earlier decreases during waterlogging. Thus, although off-season waterlogging strongly influenced the temporal dynamics of $CO_2$ fluxes, it did not alter total cumulative $CO_2$ emissions from the studied agricultural soils.

**Keywords:** soil respiration, $CO_2$ efflux, climate change, soil moisture, dissolved carbon, agricultural soil, cover crop

**Summary**

We studied how off-season waterlogging affects $CO_2$ and $CH_4$ fluxes, and dissolved carbon dynamics in two cultivated boreal mineral soils. The study was conducted with intact soil profiles in a greenhouse. Waterlogging reduced immediate $CO_2$ efflux, but $CO_2$ accumulated in porewater and was released to the atmosphere upon soil drying. Cumulative emissions remained unaltered. Our results suggest that temporary waterlogging does not suppress $CO_2$ production as much as conventionally assumed.



## 1 Introduction

Climate change is causing significant alterations in rainfall patterns globally increasing the inter- and intra-seasonal variability, and intensifying the climatic extremes (IPCC, 2023). In humid boreal regions, mild and rainy winters may lead to prolonged and more frequent soil waterlogging outside growing seasons (Mattila and Vihanto, 2024; Ruosteenoja and Jylhä, 2022). Soil

moisture is a critical environmental factor influencing soil carbon (C) fluxes (Raich and Schlesinger, 1992). Yet, substantial uncertainties remain concerning the response of soil organic matter (OM) decomposition and carbon dioxide ($CO_2$) efflux to periods of very high soil moisture (temporary waterlogging) in cultivated mineral soils (Fairbairn et al., 2023; Ghezzehei et al., 2019; Goffin et al., 2015; Huang and Hall, 2017; Moyano et al., 2018). Deepening our understanding of the relationship between soil moisture and C dynamics at or near complete soil water saturation with potentially anaerobic conditions is crucial

for improving predictions of soil-climate feedbacks in changing environmental conditions.

As opposed to conventional assumptions, recent research has shown that temporary anaerobic conditions formed upon waterlogging may even enhance rather than suppress C mineralization in mainly aerobic mineral soils (e.g. Huang et al., 2021; Huang and Hall, 2017; Liu et al., 2023; Winkler et al., 2019). In mineral soils, most of the OM is stabilized by mineral associations (Salonen et al., 2024; von Lützow et al., 2007) that provide protection against biodegradation (Cotrufo et al., 2019;

Lavallee et al., 2020). In humid soils, redox active iron(III) (Fe(III)) (hydr)oxides are particularly effective in binding reactive OM and hence, promoting C stabilization into mineral matrix (Aasano et al., 2018; Rasmussen et al., 2018; Torn et al., 1997). However, if soil saturates with water for prolonged periods and reducing conditions form in the bulk soil, (facultative) anaerobic microbes start utilizing alternative terminal electron acceptors (TEA), such as Fe(III) oxides, instead of oxygen ($O_2$), to oxidize OM (Fiedler et al., 2007; Jakobsen et al., 1981; Peters and Conrad, 1996; Ponnamperuma, 1985). Upon reduction,

the solubility of Fe increases, which leads to the disruption of Fe oxides and the release of OM associated with them, thereby making OM more vulnerable to mineralization (Chen et al., 2020; Fairbairn et al., 2023; Grybos et al., 2009; Huang et al., 2021, 2020; Huang and Hall, 2017). Indeed, although many commonly used C cycling and respiration models still predict near zero $CO_2$ fluxes at saturation (Ghezzehei et al., 2019; Moyano et al., 2013; Sulman et al., 2014), mineral soils have been shown to sustain substantial $CO_2$ production in water saturated conditions due to on-going anaerobic respiration (Fairbairn et al.,

2023; Huang et al., 2021; Huang and Hall, 2017; Liu et al., 2023; Moyano et al., 2018; Wickland and Neff, 2008; Zheng et al., 2019). The influence of anaerobic conditions on C mineralization has, however, not been uniform across studies, and it appears to depend on soil characteristics such as soil texture, the amount of active Fe oxides (Hanke et al., 2013; Liu et al., 2023; Winkler et al., 2019; Zhao et al., 2020) and the availability of labile C substrates (e.g. Winkler et al., 2019). Therefore, studies conducted with versatile soils from different climatic regimes are essential in evaluating the impact of changing rainfall patterns

on soil C dynamics.

Soil redox reactions are predominantly microbially catalyzed (Sigg, 2000; Stumm, 1995). Consequently, the availability of labile C substrates that stimulates microbial activity and $O_2$ consumption, also promotes the formation of reducing conditions



and redox transformations of Fe (Khan et al., 2019; Muhammad et al., 2021; Winkler et al., 2019). In natural and cultivated soils, plant roots are a source of readily degradable C for soil microbes (Kuzyakov, 2006; Muhammad et al., 2021). As much as half of the C photosynthetically fixed by plants can be allocated belowground (Chandregowda et al., 2023; Clemmensen et al., 2013; Keiluweit et al., 2015). In agriculture, the cultivation of overwintering cover crops has been recognized as an effective strategy to increase soil C inputs and thus, promote soil C sequestration (Kaye and Quemada, 2017; Poeplau and Don, 2015). However, it has also been observed that root exudates can stimulate decomposition of OM through priming (Bingeman et al., 1953), either directly by increasing microbial activity or indirectly by promoting the mobilization of C associated with soil minerals (Keiluweit et al., 2015; Kuzyakov et al., 2000). This, together with the changing climatic conditions, introduce complexity in assessing the net impact of cover crops on soil C dynamics (Jian et al., 2020; Poeplau and Don, 2015). To our knowledge, the impact of cover crops on C fluxes during periodic waterlogging has gained very little attention, despite the potential for root-derived C inputs to promote the mobilization of stable, mineral-associated C. Thus, it remains unknown whether and to what extent the labile C inputs from roots affect C dynamics in temporarily waterlogged conditions.

Many studies investigating the effects of anaerobic conditions on OM mobilization and $CO_2$ fluxes in mineral soils have been laboratory scale soil incubations (Bhattacharyya et al., 2018; Fairbairn et al., 2023; Huang et al., 2021, 2020; Huang and Hall, 2017; Zhao et al., 2020). The aim of our study was to take a step closer to natural field conditions by incorporating plant cover and using intact soil profiles while conducting the experiment in controlled greenhouse conditions. The data used in this study was collected from a 1.5-year soil monolith experiment which experimental procedure has been described in our preceding publication (Kronberg et al. 2024). In the present study, we focus on investigating how temporary soil waterlogging occurring during off-seasons (here determined as the time between harvest in autumn and next cultivation in spring) affects $CO_2$ and methane ($CH_4$) fluxes, and dissolved C species dynamics in two distinct cultivated mineral soil profiles (silty clay, sandy loam) with and without overwintering cover crop (Tall Fescue, *Festuca arundinacea*). Finally, an empirical model describing the dependency of soil respiration on temperature and moisture was fitted to the data to assess its ability to simulate observed $CO_2$ flux dynamics during and after waterlogging. The following hypotheses were set:

a) Temporary waterlogging increases soil DOC content due to mobilization of Fe-associated C.

b) Temporary waterlogging does not reduce $CO_2$ efflux as much as conventionally predicted based on changes in soil moisture, because ongoing anaerobic respiration can sustain substantial $CO_2$ production in mineral soils. However, despite the formation of anaerobic conditions, short-term waterlogging in cold temperature is insufficient to initiate $CH_4$ production.

c) Higher OM, clay and Fe oxide contents result in higher $CO_2$ fluxes from silty clay than from sandy loam soil.

d) Labile C compounds produced by the cover crop serve as substrates for microbes therefore enhancing microbial activity and promoting the release of Fe-associated C and $CO_2$ production in waterlogged conditions.



## 2 Material and methods

### 2.1 Soil monolith experiment

Cylindrical soil monoliths (d, 15.2 cm; h, 63 cm) were collected from two agricultural fields in Southern Finland tentatively classified as Eutric Stagnosol (60°16'23.4"N 24°56'40.6"E) and Eutric Cambisol/Mollic Umbrisol (60°49'07.3"N 23°45'54.8"E) having silty clay and sandy loam USDA texture, respectively. The monoliths were collected with a tractor-mounted soil auger in November 2020 and May 2021 (Uusitalo et al., 2012). Selected soil properties are presented in Table 1.





**Table 1.** Selected physical and chemical properties of the studied soils. The results accompanied by detailed analytical methods have been previously described in Kronberg et al. (2024).

| | Depth (cm) | Soil C & N | | | $pH_{H2O}$ | Oxides | | $\rho_b$ (kg dm$^{-3}$) | Particle size distribution | | |
| --- | --- | --- | --- | --- | --- | --- | --- | --- | --- | --- | --- |
| | | Total C (%) | WEOC (mg kg$^{-1}$) | Total N (%) | | $Al_{ox}$ (mmol kg$^{-1}$) | $Fe_{ox}$ (mmol kg$^{-1}$) | | Sand (%) | Silt (%) | Clay (%) |
| Silty clay | 0-10 | 2.6 | 48 | 0.22 | 7.1 | 62 | 112 | 1.01 | 9 | 43 | 47 |
| | 10-20 | 2.3 | 50 | 0.20 | 7.0 | 69 | 117 | 1.11 | | | |
| | 20-30 | 1.2 | 41 | 0.10 | 7.0 | 57 | 77 | 1.50 | 6 | 34 | 60 |
| | 30-40 | 0.5 | 32 | 0.05 | 7.2 | 50 | 41 | 1.53 | | | |
| | 40-50 | 0.3 | 26 | 0.03 | 7.4 | 48 | 37 | 1.48 | 3 | 39 | 58 |
| | 50-60 | 0.3 | 24 | 0.02 | 7.5 | 35 | 35 | 1.43 | | | |
| Sandy loam | 0-10 | 1.7 | 33 | 0.15 | 6.6 | 109 | 51 | 1.13 | 67 | 16 | 17 |
| | 10-20 | 1.8 | 29 | 0.15 | 6.7 | 107 | 50 | 1.26 | | | |
| | 20-30 | 1.7 | 33 | 0.15 | 6.7 | 110 | 51 | 1.40 | 71 | 15 | 14 |
| | 30-40 | 1.4 | 39 | 0.10 | 6.8 | 155 | 50 | 1.39 | | | |
| | 40-50 | 0.7 | 47 | 0.03 | 6.8 | 171 | 50 | 1.31 | 84 | 11 | 5 |
| | 50-60 | 0.5 | 45 | 0.02 | 6.8 | 152 | 34 | 1.39 | | | |

Symbols and abbreviations: WEOC, water extractable organic carbon; $Al_{ox}$, Oxalate extractable aluminum (poorly ordered oxides); $Fe_{ox}$, Oxalate extractable iron (poorly ordered oxides); $\rho_b$, soil bulk density.




The experiment was established in a greenhouse compartment at University of Helsinki in June 2021, and it is described in
detail in Kronberg et al. (2024). Briefly, 32 soil monoliths were collected from the two agricultural fields, with 16 from each
soil type. These monoliths were assigned to a factorial design with two plant treatments and two water treatments. This setup
resulted in eight unique treatment combinations (2 soil types × 2 plant treatments × 2 water treatments), each having four
treatment replicates (Fig. 1). During the 1.5 year-long experiment, three study cycles with alternating growing and off-seasons
were conducted. Here, only the results from the second and the third study cycles are presented because the first cycle was
dedicated for optimization of the experimental conditions, sampling and monitoring methods.

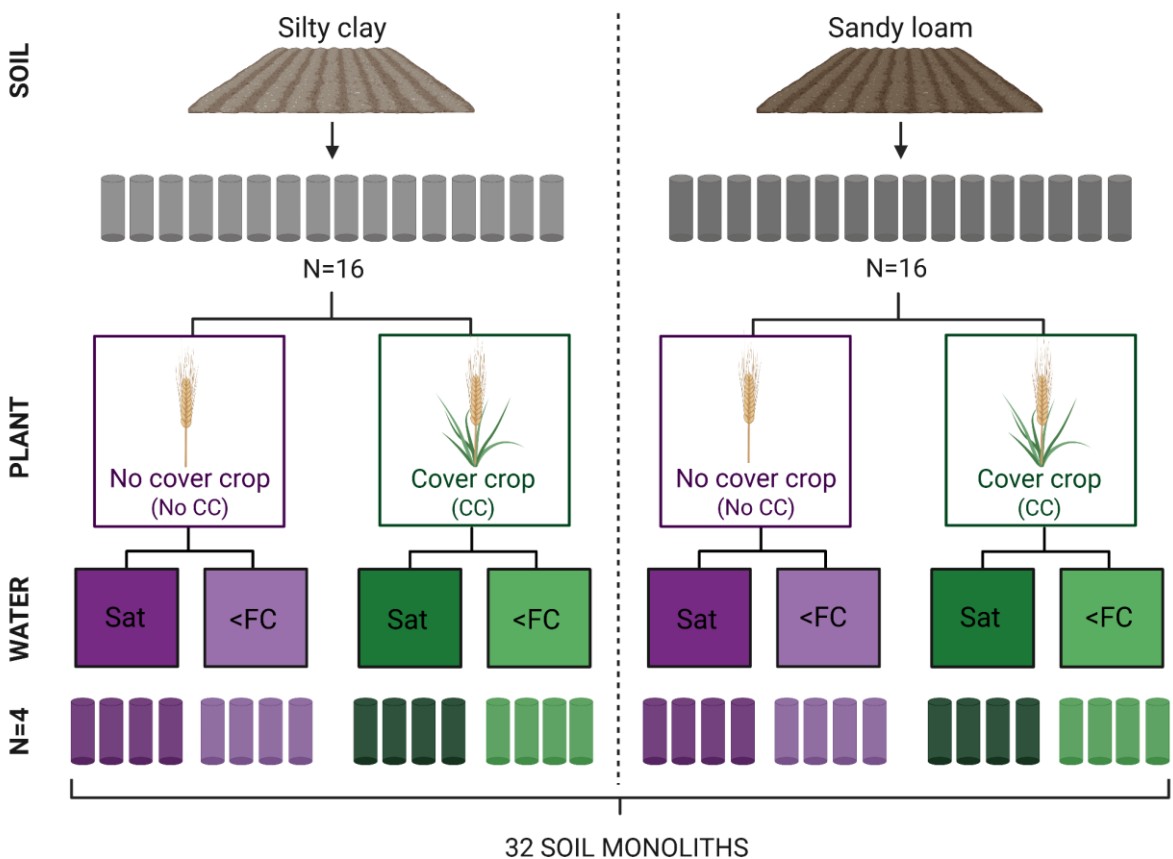

**Figure 1.** A schematic illustration of the soil monolith experiment study design. The 16 monoliths collected from each
agricultural field (silty clay, sandy loam) were assigned to two levels of plant (No CC, No cover crop; CC, Cover crop) and
water treatments (Sat, Water saturation; <FC, moisture maintained below field capacity).

Soil temperature, volumetric water content (SVWC, $m^3\ m^{-3}$) and redox potential ($E_h$) were continuously measured at 10, 30
and 50 cm soil depths in three out of four replicated monoliths (24 in total). Soil temperature and SVWC were measured with



Teros12 sensors (METER Group, USA). The soil moisture readings were calibrated to each monolith according to the gravimetric soil moisture content and bulk density determined at the end of the experiment (Kronberg et al., 2024). $E_h$ was

135 measured by platinum electrodes together with Ag/AgCl saturated KCl reference electrodes (Paleo Terra, The Netherlands). Data was logged with a Campbell datalogger (Campbell Scientific CR800 with two AM16/32B multiplexers) (Kronberg et al., 2024). The target seasonal temperatures (day/night) for the greenhouse were 20/15 °C for summer, 15/10 °C for autumn, 10/5 °C for winter and 15/10 °C for spring. The soil temperature was controlled with a separate cooler and an insulation box constructed around the soil monoliths. The realized mean soil temperature at 30 cm depth was 15.1 & 15.3 °C in summer, 10.9

& 11.7 °C in autumn, 5.7 & 7.1°C in winter and 10.9 & 10.1 °C in spring during study cycles two and three, respectively. During the waterlogging treatment the mean temperature was 6.4 °C in the second and 7.5 °C in the third cycle (Kronberg et al., 2024).

Barley (*Hordeum vulgare*) was grown in all monoliths during the growing seasons while Tall Fescue (*Festuca arundinacea*) was under sown to half of the monoliths as an overwintering cover crop. Both barley and cover crop biomass were harvested

and the dry mass was determined by drying at 105 °C for one hour and then overnight at 60 °C (Hakala et al., 2009). Afterwards, the cover crop was allowed to grow over the experimental winter and again harvested before the next cultivation. During each study cycle, all monoliths were fertilized equally using laboratory quality salts with applied nutrient levels corresponding to 90, 110 and 150 kg N ha$^{-1}$; 0, 12.5 and 25 kg P ha$^{-1}$; and 32, 25 and 100 kg K ha$^{-1}$ in cycles one, two and three, respectively. The fertilization was increased towards the end of the experiment to address nutrient deficiencies observed in barley.

Micronutrients were applied as a foliar fertilizer (YaraVita Solatrel) during the second and third study cycles (Kronberg et al., 2024).

During the growing seasons all monoliths received equal irrigation of 17–27 mm week$^{-1}$ via computer controlled Priva automated drip irrigation system (Greenhouse Environmental Control Systems, Priva BV, The Netherlands). Throughout the experiment, the monoliths were irrigated with artificial rainwater which was made according to the ionic composition of typical

rainwater in Southern Finland (Vet et al., 2014). After the harvest i.e., during the off-seasons, half of the monoliths received excessive manual irrigation until saturation was reached and water infiltration into the soil ceased (Saturation treatment). In the other half, the soil moisture was maintained below soil field capacity (<FC treatment) at ~50 % water filled pore space (Kronberg et al., 2024). Soil moisture in all monoliths was adjusted according to soil moisture readings from Teros12 sensors. The waterlogging treatment was initiated by mimicking a three-day heavy rainfall event (25 mm d$^{-1}$) that naturally occur in

Southern Finland (Uusitalo et al., 2012; Virtanen et al., 2013). The drainage outlets were closed on the third day of the heavy rainfall event (taken as day 0 of waterlogging treatment). After, these monoliths were excessively irrigated with max ~23 mm water/monolith at one irrigation event until full saturation was reached. Whereas the silty clay soil got waterlogged already during the simulated heavy rainfall event, waterlogging of the sandy loam soil took on average eight days longer due to the larger fraction of air-filled soil pores at the start of the excessive irrigation (Kronberg et al., 2024). The length of the





waterlogging treatment in cycles two and three was 54 and 50 days, respectively. The drainage was initiated by opening the drainage outlets.

## 2.2 Greenhouse gas flux measurements

Greenhouse gas fluxes ($CO_2$, $CH_4$) were measured with a closed manual chamber system including an opaque cylindrical polypropylene chamber (d, 15.2 cm; height, 20-39 cm), a non-dispersive infrared gas analyzer for $CO_2$ (Li 850, Li-cor
Environmental, USA) and a laser spectroscopy-based gas analyzer for $CH_4$ (LGR N2OM1-919, ABB Group, Switzerland). The measurement system included a controlling unit with a tablet and a Raspberry Pi (Model 3B, Raspberry Pi Ltd., UK) computer running a custom-made Python program, enabling on-line monitoring of the gas mixing ratio development in the chamber headspace. The gas analyzers were installed in parallel in the sample line to prevent problems stemming from their pumps operating at different speeds. The volume of the used chamber varied according to the plant height and ranged from
4.6 to 15.8 $dm^3$. The air inside the chamber was continuously mixed by a fan installed on the chamber sealing and the temperature was continuously monitored using a Pt-100-temperature probe and a Nokeval RMD 680 logger (Nokeval, Finland). The measurements were conducted weekly or bi-weekly during the off-seasons and bi-weekly during growing seasons.

On a measurement day each monolith was measured once for ~7 minutes. The junction between the chamber and the monolith
was sealed with plastic foil to make the chamber airtight. If any errors or leakages were observed during the closure, the measurement was repeated. The distance of the soil from the chamber edge was measured at least monthly. During the third study cycle, an empty chamber was measured before (and sometimes after) each measurement round to quantify any potential baseline fluxes. Erroneous closures, identified by fluctuations in $CO_2$ mixing ratio, were excluded from further analysis. For $CH_4$, the empty chamber flux did not deviate from zero (Wilcoxon signed-rank test, p>0.05) but the median of the empty
chamber flux for $CO_2$ was 1.25 mg $m^{-2}$ $h^{-1}$ ($Q_1$, 0.34; $Q_3$, 2.86; n, 25;). Accordingly, the empty chamber $CO_2$ flux was deducted from the measured $CO_2$ fluxes.

Gas fluxes were calculated by fitting a linear model to the gas mixing ratio data plotted against the closure time. The model fitting was conducted with a least squares method for the gas concentration change over 6 minutes. The first 30–45 seconds of the chamber closure were omitted to avoid errors caused by the system instabilities and pressure variations at closure (Pavelka
et al., 2018). Thereafter the data was visually inspected and fits with $R^2 < 0.84$ for $CO_2$ flux were classified as poor-quality data and omitted. Use of this threshold removed 0.6 % of the data.  However, closures with a $CO_2$ flux lower than three times the standard deviation of the empty chamber flux had generally lower $R^2$ values and were therefore inspected separately and omitted when when the $CO_2$ mixing ratio was fluctuating, suggesting a leak (0.4 % of the data). The fluxes using the slope term d$C/$d$t$ from the linear fit were calculated with the equation 1:



$$F = \frac{\mathrm{d}C}{\mathrm{d}t} \frac{M_{\mathrm{gas}} p V}{RTA} \cdot 3600 \qquad (1)$$

where $F$ is the gas flux (g m$^{-2}$ h$^{-1}$), $M_{\mathrm{gas}}$ molar mass of the gas (g mol$^{-1}$), $p$ air pressure (101325 Pa), $V$ chamber volume (m$^3$), $R$ ideal gas constant (8.314 J K$^{-1}$ mol$^{-1}$), $T$ air temperature (K) and $A$ is a monolith surface area (m$^2$). Positive flux indicates an efflux of gas to the atmosphere. The calculated $CO_2$ fluxes were further converted to the unit of g C m$^{-2}$ d$^{-1}$ by multiplying the fluxes by 24 h d$^{-1}$, and by dividing with the molar mass of $CO_2$ (44.01 g mol$^{-1}$) and multiplying with that of C (12.01 g mol$^{-1}$).

During the third waterlogging treatment we wanted to separate the share of the above ground cover crop respiration from the total $CO_2$ fluxes. After 43 days of waterlogging, the gas fluxes were measured twice on the same day, before and after the removal of the above ground biomass. The difference of the first and the second measurement was assumed to represent the above ground plant tissue respiration.

**2.3 Dissolved C and Fe in soil porewater**

Soil porewater samples were collected with Rhizon CSS samplers (pore size 0.15-0.2 µm, Rhizosphere Research Products B.V., The Netherlands) from three treatment replicates (monitored monoliths). At each sampling, needles (0.8x40 mm, Becton Dickinson, S.A., Spain) were inserted at the end of the samplers to connect them to 12 ml evacuated glass vials (Labco Limited, England) through rubber septa. Porewater samples were collected only from the monoliths belonging to the saturation treatment because suction was not adequate for sampling the monoliths maintained at soil moisture < FC. In the third cycle,
samples were collected only during the waterlogging treatment, whereas in the second cycle, sampling was conducted both during the treatment and after the start of the drainage, with the final sampling done 16 days after. Total dissolved C (TDC), DOC and dissolved inorganic carbon (DIC) concentrations were measured with combustion catalytic oxidation method (TOC-L, Shimadzu Corporation, Japan) and total dissolved Fe concentration was measured colorimetrically with a 1,10-phenantroline method (Hill et al., 1978, see Kronberg et al., 2024). Samples were stored at -20 °C before the analysis.

**2.4 Data analysis**

Data analysis was conducted in R studio software environment with R version 4.4.3 (R Core Team, 2025). The impact of soil type as well as plant and water treatments on average $CO_2$ fluxes were tested with linear mixed effects model in glmmTMB package (Brooks et al., 2017). The effects were tested separately for the two study cycles. Monolith and measurement round were set as a random intercept to account for the repeated measurements and the non-independency of the data. The flux data
was further divided into two temporal subgroups, during and after the water treatment (but before the next cultivation), and the model was built separately for these time intervals due to their different flux dynamics. The models were first constructed with the likelihood method (ML), and the selection of interaction terms to include in the final models was done based on Akaike´s information criteria (AIC) with stepAIC function and backwards method in MASS package (Venables and Ripley,



2002). Dropping interaction terms mostly improved the AIC apart from the period after waterlogging in cycle 3 for which the
interaction between water and plant treatment was included. All main terms (soil type, water and plant treatments) were always
included in the final model which was constructed with the restricted maximum likelihood method (REML). The assumptions
of normality and homoscedasticity of the model error terms were inspected visually in all analyses. The significance level in
all statistical analyses was set to 0.05.

Cumulative $CO_2$ fluxes were calculated by first linearly interpolating the flux measurements for each monolith with na.approx
function in zoo package (Zeileis and Grothendieck, 2005). Next, the cumulative sum was calculated starting either from crop
cultivation (beginning of the season) or from the beginning of the waterlogging treatment and calculated until next cultivation.
The cumulative $CO_2$ fluxes calculated over the entire cycles were also further normalized for cumulative above ground
biomass. The effect of soil, water and plant treatment on cumulative $CO_2$ fluxes was tested with linear model separately for
the study cycles two and three. The assumptions of normality and homoscedasticity of the model residuals were visually
evaluated by diagnostic plots. Biomass normalized cumulative $CO_2$ fluxes did not fulfill the assumption of residual normality
and hence, Box-Cox transformation was applied (Box and Cox, 1964). Tukey´s post hoc tests were further applied to test the
differences in between the treatments by using emmeans function in emmeans package (Russell, 2023). The comparison was
done among the levels of water and plant treatments for each soil type and study cycle.

$CO_2$ efflux in the monitored monoliths was modelled with an empirical equation utilizing an exponential temperature function
(Kätterer et al., 1998) combined with a parabolic WFPS function (Buysse et al., 2016; Jeanneau et al., 2020; Luo and Zhou,
2006; Moyano et al., 2013) as follows:

$$F_{CO_2} = Q_{10}^{\frac{T-10}{10}} \cdot (-a \cdot WFPS^2 + b \cdot WFPS) \tag{2}$$

$F_{CO2}$ is the soil $CO_2$ efflux (g m$^{-2}$ d$^{-1}$), $Q_{10}$ is the $CO_2$ temperature sensitivity factor, $T$ is the measured soil temperature (°C) at
10 cm depth, $WFPS$ is the water-filled pore space (%) calculated as the proportion of the measured SVWC at 10 cm from the
maximum SVWC (SVWC$_{max}$) in each monolith belonging to the saturation treatment. In <FC treatment, an average of the
SVWC$_{max}$ in the saturated monoliths in each soil type was used. In the equation, $a, b$ and $Q_{10}$ are parameters fitted for the two
soils and two plant treatments separately. Residual standard error, adjusted $R^2$ + slope from the linear fit between the modelled
and measured fluxes as well as the model residuals were used to evaluate the model performance. The constructed models
were used to calculate the daily $CO_2$ flux for each monitored monolith with daily average values of soil $T$ and $WFPS$ (n=3).
The unit of the modelled flux was further converted to g $CO_2$-C m$^2$ d$^{-1}$. Obtained fluxes were also used to calculate the
cumulative $CO_2$ fluxes similarly than with measured values starting from the beginning of the water treatment until next
cultivation. The difference in cumulative fluxes calculated based on measured vs. modelled values was tested with pairwise t-
test.



To study the relationship between dissolved C and Fe concentrations in soil porewater, repeated measures correlation analysis, where monolith was set as a participant/subject, was performed with rmcorr package (Bakdash & Marusich, 2024). Then, the concentrations of dissolved C species in soil pore water ($c$, mg l$^{-1}$) were further converted to the amount of C per liter soil (mg l$^{-1}$ soil) by multiplying the pore water concentration by the daily average SVWC (dm$^3$ dm$^{-3}$) in each 20 cm soil layer. This was done to eliminate the apparent effect of SVWC on soil C contents in changing moisture conditions. In each monolith, the amount of accumulated DIC, i.e., the change in soil DIC storage ($\Delta DIC$, mg C m$^{-2}$), during waterlogging was calculated with the equation 3. This involved summing the change in DIC content from the three 20 cm soil layers (0–20 cm, 20–40 cm, and 40–60 cm) multiplied by their volume ($V$, 3.54 dm³), and then dividing the total by the monolith surface area ($A$, 0.018 m²).

$$\Delta DIC = \frac{\sum_{Layer=1}^{3} V \cdot (\theta_{end} \cdot c_{end} - \theta_{start} \cdot c_{start})_{Layer}}{A} \qquad (3)$$

In the equation, $c_{end}$ and $c_{start}$ are the soil porewater DIC concentrations (mg l$^{-1}$) at the end and at the first sampling of the waterlogging period, respectively, and $\theta_{end}$ and $\theta_{start}$ are the corresponding volumetric soil water contents. In cases where the $c_{start}$ was not available for a specific soil monolith (no porewater obtained), the average of the first sampling round in each soil type and depth calculated based on available monoliths was used. The equilibrium porewater $CO_2$ concentration at the start was assumed to be similar within the same soil type and depth. The concentration was also assumed to be similar within each 20 cm soil layer around the sampling depth.

 Characteristic of Finnish soils, neither of the studied soils contained significant amounts of carbonates (verified with 0.2 M HCl, "Fizz test") and thus, $\Delta DIC$ was assumed to consist mainly of dissolved $CO_2$ deriving from soil respiration. Thus, the total amount of respired $CO_2$ during the waterlogging was estimated by adding the $\Delta DIC$ to the cumulative $CO_2$ fluxes from each monolith. Only the waterlogging period from which porewater samples were collected was considered (Cycle 2: waterlogging days 0-49; cycle 3: days 7-45). In <FC treatment, only the cumulative $CO_2$ flux was considered because no porewater was obtained and the $\Delta DIC$ was assumed to be negligible because of the gas exchange with the atmosphere. The effect of time (days), soil type, plant treatment and depth on the concentration of TDC, DIC, DOC and total $CO_2$ production were tested with linear mixed effects model and nlme package (Pinheiro et al., 2023). A stepwise (backwards) model selection was performed with stepAIC function in MASS package similarly than with greenhouse gas fluxes (Venables and Ripley, 2002). The effect of waterlogging on total respired/produced $CO_2$ (in comparison to the <FC treatment) was further tested by pairwise comparisons by emmeams function (Russell, 2023). The comparison was done separately for the two cycles within soil type and plant treatment.





# 3 Results

## 3.1 Measured $CO_2$ and $CH_4$ fluxes

During the waterlogging treatment the $CO_2$ fluxes were significantly ($p<0.01$) lower in the saturated soil monoliths than in the control monoliths kept below FC (Fig. 2, Table A1). $CO_2$ fluxes from the saturated monoliths without cover crops were mostly around (cycle 2) or below (cycle 3) $0.5$ g C m$^{-2}$ d$^{-1}$, and fluxes in the monoliths with cover crop were significantly higher (+ $0.18$ and + $0.87$ g C m$^{-2}$ d$^{-1}$ in cycles 2 and 3, respectively. Table A1). Upon drainage, the fluxes from the saturated monoliths

increased to significantly higher level than in the <FC treatment ($p<0.001$, Fig. 2). The observed $CO_2$ pulse upon drainage was steeper in sandy loam compared to the silty clay soil (Fig. 2). After the $CO_2$ pulse in cycle 2, the differences between the two water treatments levelled out within two weeks in sandy loam soil reaching a level of ~2 g C m$^{-2}$ d$^{-1}$ in No CC and ~3.5 g C m$^{-2}$ d$^{-1}$ in CC treatment. In silty clay soil, $CO_2$ fluxes from the previously saturated monoliths remained higher until the next cultivation which was four weeks after drainage. Soil $CH_4$ fluxes were negligible or slightly negative in all treatments during

both water saturation and drainage (data not shown).

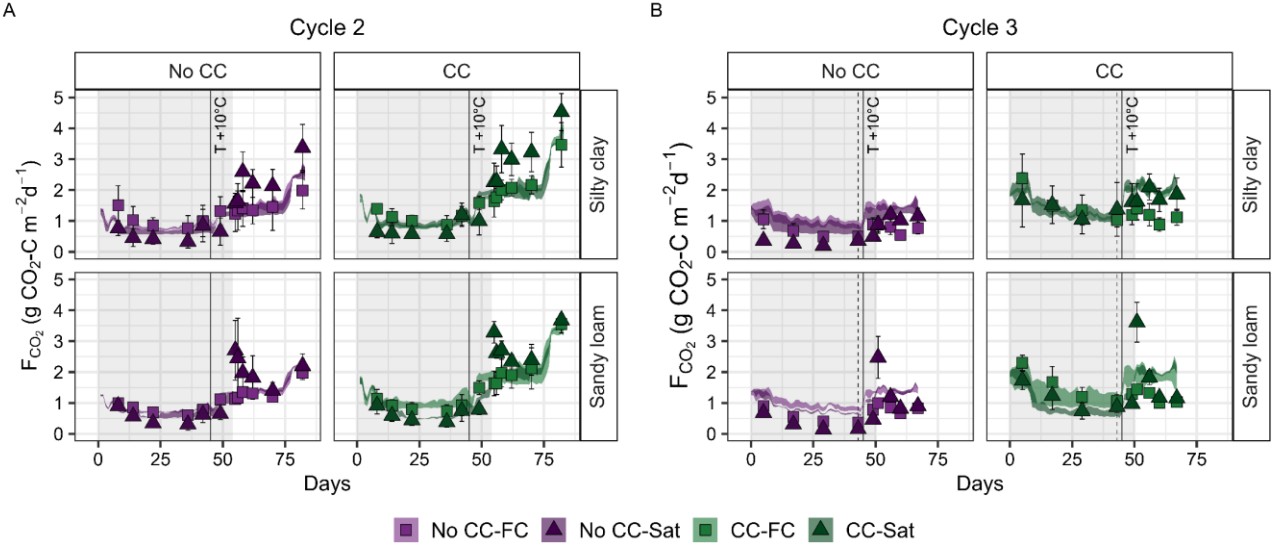

**Figure 2**. Measured (mean ± standard deviation, n=4) and modelled (shaded ribbons, range, n=3, only monitored monoliths used in modelling) $CO_2$ efflux during and after waterlogging until the next cultivation in cycle 2 (A) and until the end of the experiment in cycle 3 (B) in silty clay and sandy loam soil with (CC) and without cover crops (No CC) in the two water

treatments (FC, < field capacity; Sat, Water saturation). The gray shaded area represents the duration of the waterlogging, the vertical solid line the day when the soil temperature was increased from 5 °C to 10 °C, and the dashed line the day of cover crop biomass removal.



Temporal $CO_2$ flux dynamics varied slightly between the cycles two and three. In the third cycle, $CO_2$ fluxes from the saturated monoliths with the cover crop, especially in silty clay soil, started to increase already before the initiation of drainage coinciding with the removal of the above ground cover crop biomass. This earlier increase in $CO_2$ fluxes in the cover crop treatment reduced the difference in the average $CO_2$ fluxes between the waterlogged and the control treatment (Table A1). However, the water saturation still resulted in significantly lower $CO_2$ fluxes in both plant treatments, as already indicated.

Waterlogging did not have a statistically significant impact on off-season cumulative $CO_2$ fluxes in either of the study cycles, plant treatments or soils (Fig. 3). In the third cycle, however, the mean cumulative $CO_2$ fluxes in the saturated soils without the cover crop (silty clay $34 \pm 5.0$ g C m$^{-2}$, sandy loam $40 \pm 4.9$ g C m$^{-2}$) were slightly lower than in the <FC treatment (silty clay $49 \pm 11.3$ g C m$^{-2}$, sandy loam $46 \pm 7.0$ g C m$^{-2}$). In turn, according to the linear model, the cover crop significantly increased the cumulative fluxes ($p<0.01$), while the soil type did not have a significant impact on them ($p=0.06$ or $0.87$ in cycle two and three, respectively). The difference between the two plant treatments was larger in the third than in the second cycle (Fig. 3) and the post hoc tests revealed that in the second cycle the difference between the CC and No CC monoliths was barely insignificant in both water treatments in sandy loam soil ($p=0.08/0.09$) and in the <FC treatment in silty clay ($p=0.09$). The treatment-wise pattern in cumulative $CO_2$ fluxes over the entire study cycles (growing and off-seasons combined) remained consistent with those calculated for the off-seasons alone, but only when the fluxes were normalized for above-ground biomass (Fig. 3). Without this normalization, cumulative $CO_2$ fluxes over the entire cycle did not differ in the two plant treatments (data not shown), as opposed to the differences observed during the off-seasons.



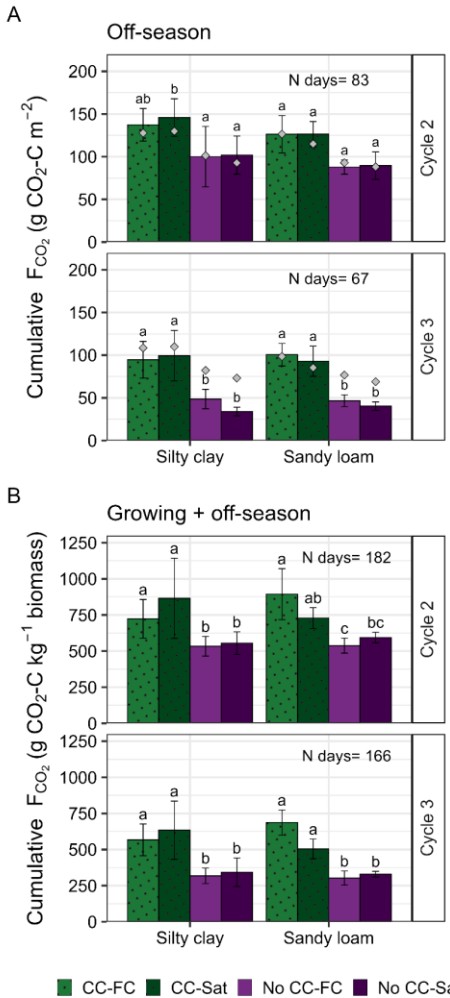

**Figure 3**. Cumulative $CO_2$ fluxes (as g $CO_2$-C) calculated for the off-season starting from the initiation of the waterlogging treatment until the next cultivation (Cycle 2) or until the end of the experiment (Cycle 3) (A), and for the entire study cycle (Cycle 2, Cycle 3) normalized to the cumulative above ground biomass (kg$^{-1}$ aboveground biomass) (B). Mean ± standard deviation (n=4) of the replicated monoliths (CC-FC, Cover crop & <field capacity; CC-Sat, Cover crop & Water saturation; No CC-FC, No cover crop & <field capacity; No CC-Sat, No cover crop & Water saturation). Gray squares illustrate the mean (n=3) cumulative flux from the empirical temperature and moisture dependency model. *N days* represents the number of days included in the calculation of the cumulative sums, and lower-case letters the statistical differences within each soil type and cycle.

In the <FC treatment, the amount of above ground cover crop respiration, determined as the difference between the measured fluxes before and after above ground biomass removal, was on average $0.6 \pm 0.42$ g $CO_2$ m$^{-2}$ h$^{-1}$ and $0.5 \pm 0.55$ g $CO_2$ m$^{-2}$ h$^{-1}$





in silty clay and sandy loam monoliths, respectively. This represented 15% and 12% of the total $CO_2$ flux in silty clay and sandy loam, respectively. However, the averages were not statistically different from zero in either of the soils. Instead of a decrease, the above ground biomass removal induced an increase in $CO_2$ fluxes from saturated monoliths. In silty clay soil, the fluxes increased on average $0.6 \pm 0.88$ g m$^{-2}$ d$^{-1}$ while the increase in sandy loam soil was modest, on average only $0.1 \pm 0.59$

330   mg m$^{-2}$ d$^{-1}$. However, because of the large deviation between the replicate monoliths, the increases did not deviate from zero statistically significantly.

### 3.2 $CO_2$ efflux modelled with the empirical model

In the empirical temperature and moisture dependency model fit to our data, soil temperature dominated over moisture as the predictor for $CO_2$ efflux as indicated by the high $Q_{10}$ and low $a$ and $b$ parameter values (Table 2). The $Q_{10}$ values in the cover

crop treatment (silty clay=4.79; sandy loam=3.79) were higher than the corresponding values without the cover crop (3.49 and 3.31, respectively) (Table 2). The modelled fluxes were very similar in the waterlogged and in <FC treatment especially in silty clay soil (Fig. 2). The $CO_2$ flux dynamics and response to soil moisture were better captured (Fig. 2) and the model performance was overall better in sandy loam soil which is illustrated by the smaller RSE of the model and a larger $R^2$ of the linear model fitted for the predicted vs. measured data than in silty clay soil (Table 2). In both soils, model performed better

with the cover crop than without the cover crop as indicated by higher $R^2$ (Table 2).

There was a clear time wise trend in model residuals especially during the second cycle in the monoliths without cover crops (Fig. B1). Towards the end of the waterlogging, the model overestimated the fluxes in the saturation treatment. In turn, the model could not reproduce the $CO_2$ peak measured after the initiation of drainage which led to underestimation of the fluxes (positive residuals) upon drainage in the monoliths belonging to the saturation treatment (Fig. 2, Fig B1). With the cover crop,

the residuals were generally more randomly distributed during the waterlogging treatment but peaked after drainage similarly than without cover crops. During the third waterlogging, the model overestimated the $CO_2$ fluxes in the monoliths without the cover crop (Fig. 2). This overestimation could be fixed by fitting the model separately for the two study cycles which shifted down the predicted values during waterlogging and resulted in higher $Q_{10}$ values especially in the monoliths without cover crops (data not shown). Yet, similar increase in model residuals was still observed during the third than during the second

drainage in all treatments except for the silty clay monoliths with the cover crop.

In cycle two, the cumulative $CO_2$ efflux calculated with modelled data was in a good agreement with those calculated from the measured data with no statistically significant difference between the two (Fig. 3, Table A3). During cycle three, the same was true only for the monoliths with the cover crop. Without cover crops, cumulative $CO_2$ efflux obtained from the modelled $CO_2$ fluxes was significantly higher in both soils and water treatments (Fig. 3, Table A3).




**Table 2.** Results of the temperature (T) and water filled pore space (WFPS) dependency model fit to $CO_2$ flux data from the monitored monoliths during the off-seasons. Also, the adjusted $R^2$, slope and intercept from the linear fit between the modelled and measured fluxes, used to evaluate model performance, are presented.

| | | $FCO_2$ ~ T, WFPS | | | | | Predicted ~ Measured | | |
|---|---|---|---|---|---|---|---|---|---|
| | | RSE | Q10 | a | b | N | Intercept | Slope | $R^2$ |
| No cover crop | Silty clay | 2.24 | 3.49 | 0.0009 | 0.13 | 145 | 2.73 | 0.38 | 0.39 |
| | Sandy loam | 1.88 | 3.31 | 0.0010 | 0.13 | 150 | 2.42 | 0.41 | 0.45 |
| Cover crop | Silty clay | 2.22 | 4.79 | 0.0008 | 0.14 | 150 | 2.58 | 0.60 | 0.59 |
| | Sandy loam | 1.93 | 3.79 | 0.0018 | 0.21 | 150 | 2.14 | 0.65 | 0.66 |

RSE=Residual standard error of the model (g C $m^{-2}$ $d^{-1}$), $Q_{10}$=fitted $CO_2$ flux temperature sensitivity parameter, a & b = fitted model parameters, N=number of observations, Intercept=intercept of the linear regression fitted for modelled and measured $CO_2$ fluxes, Slope=slope of the linear regression, $R^2$=adjusted coefficient of determination of the linear regression.

### 3.3 Dissolved C dynamics in porewater

Total dissolved C (TDC) content increased during waterlogging at 10 and 30 cm depths in both soils and under both plant treatments (Fig. 4). The observed increase was attributed to the increased dissolved inorganic carbon (DIC) content while dissolved organic carbon (DOC) content remained relatively unchanged. The increases in TDC and DIC contents during waterlogging at 30 cm depth were more pronounced in sandy loam than in silty clay, whereas especially in the second cycle the increase in DIC content in the topsoil was steeper in silty clay (Table A5 & S6, Fig. 4). In sandy loam, waterlogging induced an increase in DIC content also at 50 cm depth; there the increase also persisted after the initiation of drainage. The continuing increase after drainage was also observed in silty clay soil at 30 cm depth for DIC and TDC contents. In the topsoil the DIC content started to decrease soon after drainage in both soils returning to levels observed at the start of the waterlogging period (Fig. 4).

In the topsoil the contents of all dissolved C species (TDC, DIC, DOC) were significantly higher in silty clay than in sandy loam soil as illustrated by the significant soil term (for TDC, DIC, DOC) as well as the significant interaction of soil and waterlogging days (for TDC, DIC) in the mixed effects models (Table A4-6). In sandy loam, DOC content was generally higher in deeper soil layers than in the topsoil while the opposite was observed in silty clay soil (Fig. 4). Cover crops increased the TDC content in the silty clay topsoil while in sandy loam the increase was more pronounced at 30 cm depth (Fig. 4). In the third cycle, the average DOC content at 50 cm depth was significantly higher in the monoliths with the cover crop than without in sandy loam soil (p=0.01) (Fig. B2), while there was no statistically significant difference during the second cycle (Table A6).



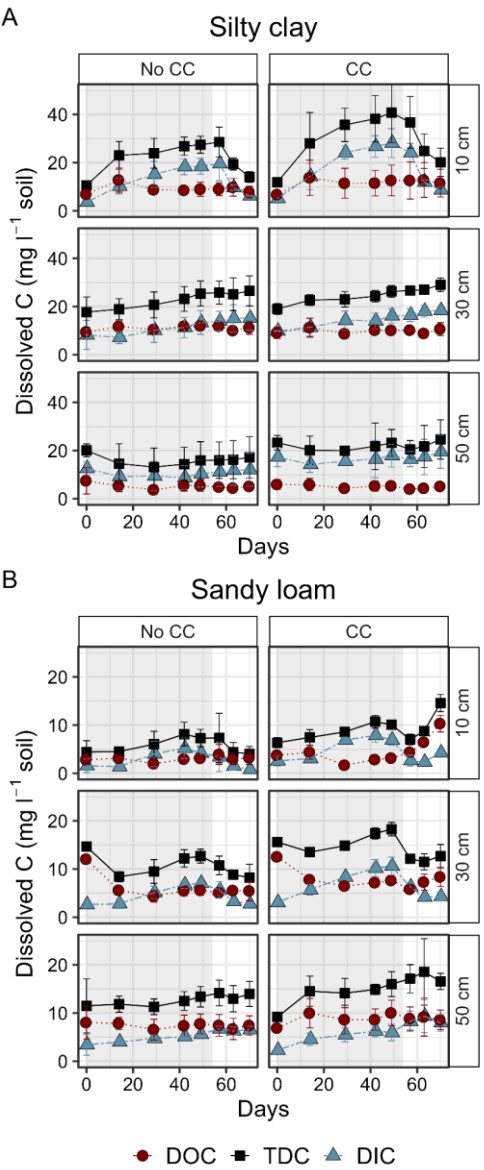

**Figure 4.** Timeseries of dissolved organic carbon (DOC), inorganic carbon (DIC) and total dissolved carbon (TDC) contents as mg C l$^{-1}$ soil (mean ± standard deviation, n=3) during (shaded area) and after waterlogging in silty clay (A) and sandy loam (B) soil profiles with and without cover crops (No CC, No cover crops; CC = Cover crops) in cycle two. Days on the x-axis represent the days since the beginning of waterlogging. Note the differing scales on y-axis in the two soils. Data from cycle three is presented in supplementary Fig. S1.



During the second off-season, porewater TDC concentration correlated with dissolved Fe concentration at 10 and 30 depths in silty clay soil both with (10 cm: r= 0.51, p=0.023; 30 cm: r=0.58, p=0.005) and without the cover crop (10 cm: r= 0.46, p=0.049; 30 cm: r=0.58, p=0.006), and in sandy loam with the cover crop (10 cm: r= 0.58, p=0.009; 30 cm: r=0.59, p=0.07) (Table A9). In silty clay soil, the correlation with dissolved concentration Fe was attributed to the changes in DIC concentration in both plant treatments (Fig. 5A), whereas in sandy loam Fe concentration correlated with DOC instead, however, only in the

topsoil (Fig. 5D). During the third cycle the porewater C and Fe concentrations were only measured during the waterlogging (excluding the drainage phase) and then, the DOC concentration did not exhibit positive correlation with Fe in either soil at any depth (Table A9). During this cycle, the correlation between DIC and Fe concentrations, in turn, was significant in both soils with the cover crop at 30 cm (silty clay: r=0.88, p<0.001; sandy loam: r=0.71, p=0.022) and in silty clay at 10 cm with the cover crop (r=0.75, p=0.013) (Table A9).



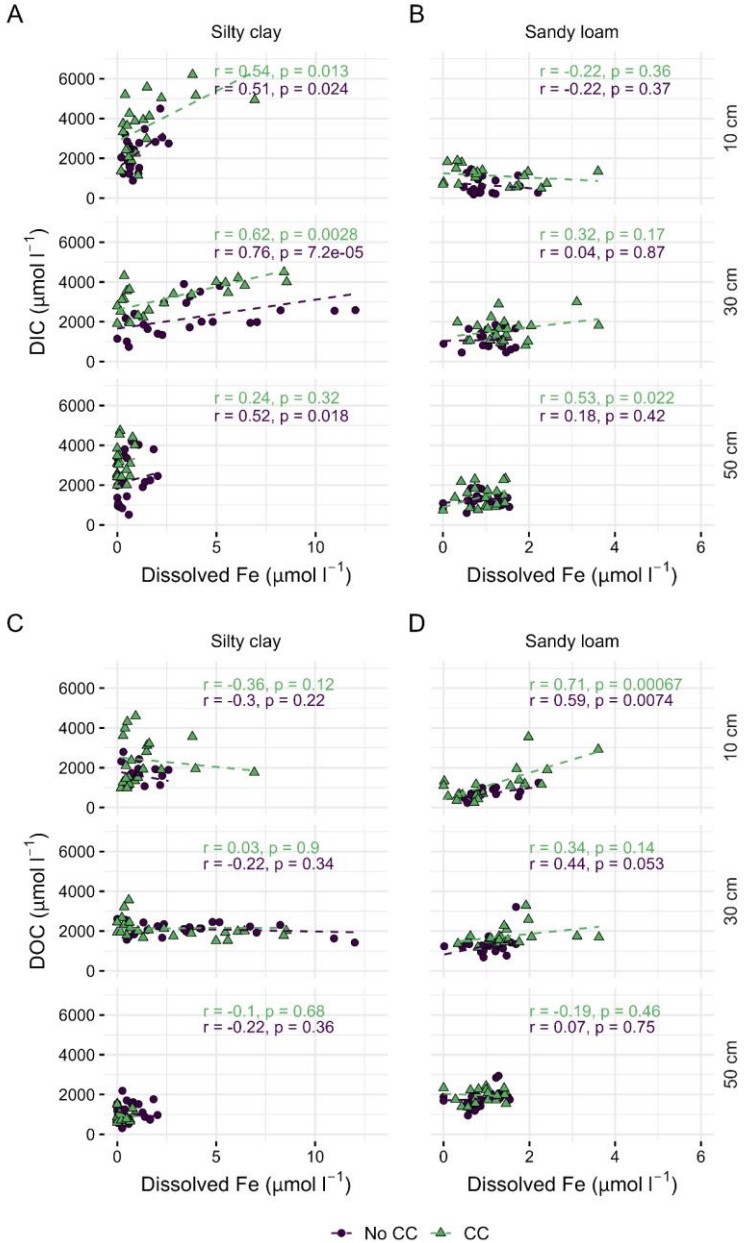

**Figure 5.** Relationship between porewater dissolved Fe and dissolved inorganic carbon (DIC) (A-B) and dissolved organic carbon (DOC) (C-D) in the soils belonging to the saturation treatment, and with (CC) and without the cover crop (No CC) at different depths. The regression lines, and the correlation coefficients (r) together with the corresponding p-values were obtained from repeated measures correlation analysis. Observe the differing x-axis scales in the two soils.





**3.4 Total CO$_2$ production as the sum of cumulative CO$_2$ fluxes and ΔDIC**

Results from the linear model showed that waterlogging decreased the total CO$_2$ production (cumulative CO$_2$ flux + ΔDIC) statistically significantly during the second (p=0.002) but not during the third (p=0.10) waterlogging (Fig. 6, Table A7). However, according to the pairwise comparisons, the means of the two water treatments within each soil type and plant treatment were only statistically significantly different in silty clay soil in cycle two without the cover crop (Fig. 6, Table A8). In the third cycle CO$_2$ production was statistically significantly higher with the cover crop than without (+35 g C m$^{-2}$, p<0.01) while there was no difference in the second cycle (p=0.1). Soil type was significant in cycle two where the total CO$_2$ production during waterlogging was lower in sandy loam than in silty clay (-8.3 g C m$^{-2}$, p=0.05) (Fig. 6). Soil type was not significant in cycle three.

The amount of the accumulated DIC in the saturated monoliths corresponded to on average 6–29% of the total CO$_2$ production during waterlogging. In the second cycle, the percentage was on average 11.5 ± 5.7% (No CC) and 13.5 ± 0.4% (CC) in silty clay, and 6.0 ± 1.7% (No CC) and 8.8 ± 3.9% (CC) in sandy loam. In the third cycle, the percentage was larger in No CC treatment (silty clay 28.8 ± 2.5%, sandy loam 19.8 ± 2.6%) but it remained at a similar level in the CC treatment (silty clay 9.4 ± 1.4%, sandy loam 8.9 ± 2.4%).



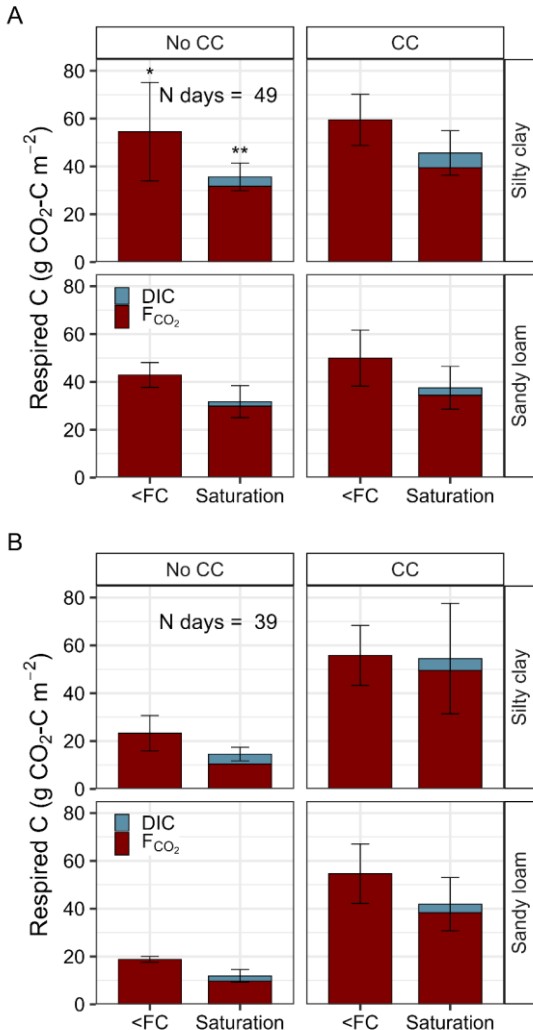

**Figure 6.** Total respired/mineralized $CO_2$-C as the sum of cumulative $CO_2$ flux and accumulated DIC in soil pore water during the second (A) and third (B) waterlogging in silty clay and sandy loam soil with (CC) and without (No CC) cover crops. Statistically significant differences between the two water treatments are marked with asterisks.

## 4 Discussion

### 4.1 Impact of waterlogging on measured $CO_2$ and $CH_4$ fluxes

We studied the impact of periodic off-season waterlogging on soil C dissolution, and $CO_2$ and $CH_4$ emissions in a controlled mesocosm study with intact soil profiles with and without Tall Fescue as a cover crop. In our experiment, temporary soil





waterlogging lasting ~50 days at an average soil temperature of ~7 °C did not significantly alter cumulative $CO_2$ emissions calculated over waterlogging and subsequent drainage period. Our results align with findings from several incubation studies suggesting that temporary anaerobic conditions may exert only a minor impact on C decomposition and cumulative greenhouse

gas emissions, contrary to conventional model assumptions (e.g. Bhattacharyya et al., 2018; Hanke et al., 2013; Huang et al. 2021; Tete et al., 2015; Winkler et al., 2019). In some studies, anaerobic conditions have even been shown to enhance C mineralization due to mobilization of mineral-associated C (Huang et al., 2021; Huang and Hall, 2017; Jia et al., 2020; Liu et al., 2023; Possinger et al., 2020; Winkler et al., 2019). This enhancement has been observed particularly under prolonged and/or fluctuating anaerobiosis (Huang et al., 2017; Huang et al., 2021, Possinger et al., 2020), in soils where a substantial

portion of C has been stabilized by mineral associations (Liu et al., 2023), and when labile C substrates have facilitated microbial Fe reduction (Winkler et al. 2019). Thus, based on previous studies, the response of soil C to anaerobic conditions appears to be soil dependent.

Contrary to our hypothesis, the cumulative $CO_2$ fluxes, and their response to waterlogging, were similar in the two soils, despite differences in texture and Fe oxide content. This was unexpected also because the higher DOC and DIC contents measured in

the silty clay than in the sandy loam topsoil during waterlogging implied a higher mineralization rate. The dissolved C dynamics in deeper soil layers help to explain the lack of difference in the aggregated $CO_2$ fluxes from the whole profile. The marked increase in DIC content at 30 cm depth in the sandy loam soil, and a more modest increase in the silty clay suggests that while $CO_2$ production was lower in the sandy loam topsoil compared to silty clay, higher microbial activity at 20–40 cm depth in sandy loam increased the total respiration per area, thereby narrowing the difference in cumulative $CO_2$ emissions

between the two soils. Most studies on soil C dynamics focus exclusively on changes in the top 0–20 cm soil layer (Poeplau and Don, 2015). However, it has been demonstrated that examining the entire soil profile can yield conclusions that greatly differ from those based solely on topsoil analysis (Tautges et al., 2019). Thus, our findings emphasize that C dynamics in deeper soil layers may play a significant role in overall C dynamics and should therefore not be overlooked in future studies.

Although the cumulative $CO_2$ fluxes were not significantly affected by waterlogging in our study, the temporal flux dynamics

differed between the water saturated monoliths and those maintained at <FC moisture. The reduced $CO_2$ fluxes during waterlogging were compensated for by the higher fluxes after the drainage in both soils. Our results align with those of Huang et al. (2021) and Tete et al. (2015) who conducted laboratory incubations with mineral soils collected from agricultural and/or forested sites. In both studies, reduced decomposition during anaerobic period(s) was compensated by increased $CO_2$ efflux during the following aerobic period(s) in weekly to monthly timescale. Based on these observations, during transiently

anaerobic events, most $CO_2$ fluxes tend to occur during the subsequent drainage/aerobic phase (Huang et al., 2021; Tete et al., 2015) which makes the calculation of cumulative emissions sensitive to the chosen observation period. Thus, our results underscore the critical role of the following drying period in determining the overall impacts of waterlogging on soil C budget in fluctuating moisture conditions.



As hypothesized, waterlogging did not induce $CH_4$ production in either soil. This aligns with the measured soil reduction-
oxidation (redox) potentials, whose lowest average values were -110 mV and -60 mV in silty clay and sandy loam soil,
respectively (Kronberg et al., 2024). This indicates that conditions did not reach the strongly reducing levels associated with
$CH_4$ production (Reddy et al., 2022). Methanogenesis is generally believed to occur only after alternative terminal electron
acceptors (TEAs) with higher energy yields, such as $NO_3^-$, Fe(III), $SO_4^{2-}$, have been utilized (Burgin and Loecke, 2023; Peters
and Conrad, 1996; Reddy et al., 2022). Thus, the on-going Fe(III) reduction, as indicated by the slightly increasing dissolved
Fe concentrations during waterlogging in the clay soil pore water together with the accompanied $E_h$ values associated with Fe
reduction (Kronberg et al., 2024), likely inhibited methanogenesis in our experiment.

## 4.2 Could the simple empirical model capture altered $CO_2$ flux dynamics during and after waterlogging?

In line with recent studies (e.g. Fairbairn et al. 2023; Huang et al. 2021), the conventional $CO_2$ efflux model was unable to
simulate flux dynamics during and after waterlogging. Yet, it quite accurately predicted the cumulative $CO_2$ fluxes in the
second study cycle. In neither of the soils could the model capture the pulse in $CO_2$ fluxes after the initiation of the drainage.
Overall, the model performed better in sandy loam soil, where the soil moisture differences between the two water treatments
were more pronounced, and where moisture changes upon drainage occurred faster due to the soil's lower water retention and
better water conductivity compared to silty clay. In turn, the modelled fluxes were insensitive to soil moisture in silty clay
where the moisture changes upon drainage were slower and smaller.

The $Q_{10}$ parameter in the empirical model expresses the sensitivity of soil respiration to soil temperature (Davidson et al.,
2006; Kätterer et al., 1998). The $Q_{10}$ values obtained from the model fit to our data were high (3.31–4.79) compared to values
reported by Kätterer et al. (1998) (1.35–2.88) or obtained from in situ field measurements from mineral agricultural soils in
Finland (1.7–2.3) (Koizumi et al., 1999; Lind et al., 2016; Lohila et al., 2003). However, the values from the field studies have
been recorded during growing seasons while in our experiment, covering growing season and off-season, the lower temperature
could be expected to result in slightly higher $Q_{10}$ values (Davidson et al., 2006; Maier et al., 2011). Still, according to Davidson
et al. (2006), "apparent" $Q_{10}$ values above 2.5 usually include effects from other processes than direct temperature sensitivity
of microbial respiration. For example, it has been shown that $Q_{10}$ tends to increase with increasing soil moisture due to
increased substrate availability to microbes (Buysse et al., 2016; Zhou et al., 2014). While this could partly explain the high
$Q_{10}$ values in our study, particularly given that soil moisture in our columns was high compared to the conditions in the above-
mentioned field studies, we believe the primary reason was that the increase in soil temperature coincided with a decrease in
soil moisture that, in turn, induced a peak in $CO_2$ fluxes during spring. As a result, part of the moisture effect on respiration
was likely captured within the $Q_{10}$ estimates. We acknowledge that, in this regard, our experimental design may not have been
ideal for modelling purposes.




### 4.3 Belowground C dynamics during waterlogging

**4.3.1 Waterlogging causes a discrepancy between respired CO₂ and measured flux**

Increasing DIC content, reflecting the amount of soil $CO_2$ in soil pores, revealed that during waterlogging, C mineralization continued at a higher rate than could have been estimated solely based on $CO_2$ flux measurements. Our observations demonstrate that, contrary to convention, momentary soil $CO_2$ efflux should not be considered equivalent to soil respiration as the two are decoupled in periodic high soil moisture events (Maier et al., 2011; Ryan and Law, 2005; Sánchez-Cañete et al.,
2018). In the long term, the two can usually be considered equal as all respired $CO_2$ will eventually be released into the atmosphere (Maier et al., 2011). However, momentary fluxes can significantly deviate from respired $CO_2$ (Bond-Lamberty et al., 2024; Maier et al., 2011, 2010; Sánchez-Cañete et al., 2018). Previous research has shown that over one third of the respired $CO_2$ can get temporally stored in soil during heavy rainfall events (Maier et al., 2011; Sánchez-Cañete et al., 2018). In our experiment, accumulated $CO_2$ (ΔDIC) during waterlogging was at most ~30% of the total $CO_2$ production (sum of cumulative
$CO_2$ fluxes and ΔDIC) which is in a good agreement with previously reported values (Maier et al., 2011; Sánchez-Cañete et al., 2018). The deviations of respired $CO_2$ from momentary fluxes result from changes in diffusive transport of gases within the soil profile, being of great importance under fluctuating soil moisture conditions. When soil pore space fills with water, diffusive movement of gases is severely hindered because gas diffusion in water is 10000 times slower than in air (Greenway et al., 2006). Accordingly, respired $CO_2$ gets entrapped within soil pores resulting in discrepancy between produced $CO_2$ and
measured soil $CO_2$ efflux (Jeanneau et al., 2020; Maier et al., 2011, 2010; Sánchez-Cañete et al., 2018).

To account for the difference in measured $CO_2$ efflux and soil respiration, Maier et al. (2011, 2010) and Hirsch et al. (2004) have proposed an incorporation of a storage flux term in soil moisture dependency models. They defined the storage flux as the flux resulting from changes in the amount of $CO_2$ stored in soil (Hirsch et al., 2004; Maier et al., 2011, 2010). Our attempt to model the $CO_2$ efflux with the simple empirical temperature and soil moisture dependency model supports this idea. The
505 observed timewise trend in model residuals indicates that some underlying processes affecting measured $CO_2$ fluxes were not captured by the model. Whereas the model was able to estimate the cumulative $CO_2$ emissions during the second cycle, it overestimated the emissions during waterlogging and underestimated them upon drainage. Thus, we think that the model was unable to reproduce the $CO_2$ pulse upon drainage because the pulse did not represent momentary soil respiration but was largely a consequence of the release of previously respired $CO_2$ to the atmosphere. Maier et al. (2011) showed that soil respiration,
taken as a composite of measured and storage flux, had a stronger dependence on soil moisture than the measured efflux alone. We therefore believe that incorporating the storage flux could have also improved model predictions in our study. However, a robust incorporation of such term would have required better spatial resolution of the DIC data across soil profiles.



### 4.3.2 Processes contributing to the increase in dissolved C content

We hypothesized that soil waterlogging would promote mobilization of the Fe associated C and therefore result in an increase
in soil DOC content, as commonly observed in incubation studies (e.g. Chen et al., 2020; Huang and Hall, 2017; Pan et al., 2016; Winkler et al., 2019). As opposed to our hypothesis, soil DOC content did not increase in either soil during waterlogging; the only clear increase was observed in sandy loam topsoil after drainage, likely due to enhanced OM decomposition and plant root exudation. However, TDC content increased during waterlogging, suggesting that an immediate microbial mineralization of DOC to $CO_2$ may have masked any increases in DOC content, with the effect instead becoming apparent as an increase in
DIC (Fairbairn et al., 2023). In addition to mobilization of mineral associated C, in the presence of plant cover, the accumulating $CO_2$ (DIC) may have also originated from root respiration or from the decay of root litter (Kuzyakov, 2006). The above-mentioned processes and their contributions to increased TDC/DIC are discussed next.

Our results suggest that the contribution of Fe dissolution and the mobilization of associated C in the overall increase in TDC/DIC content was likely small. Waterlogging appeared to induce only slight reductive Fe dissolution since the Fe
concentrations in porewater remained low throughout the study (max ~10 µmol l$^{-1}$) (Kronberg et al., 2024). In silty clay topsoil, for instance, the highest average DIC concentration during the second cycle (~3 mmol l$^{-1}$) was three orders of magnitude higher than that of Fe (~2 µmol l$^{-1}$). Thus, although soil DIC concentration correlated with Fe concentration during waterlogging, particularly in silty clay soil, the dissolution of Fe-OC alone can not explain the observed increases in DIC/TDC. Vice versa, the higher dissolved C content, reflecting soil microbial/rhizospheric activity, may have rather enhanced Fe dissolution slightly,
as also seen in previous studies (e.g. Winkler et al. 2019). Overall, substrate availability was likely not substantially enhanced by waterlogging, as cumulative $CO_2$ fluxes did not increase relative to the <FC treatment.

Root respiration, including the activity of roots and closely associated microbes, can represent between 10 and 90% of the total soil respiration (Hanson et al., 2000). In our experiment DIC increased only slightly more with the cover crop than without despite the ~10 times higher root biomass at 0–20 cm depth with the cover crop (Kronberg et al., 2024). Thus, if a substantial
fraction of the accumulated DIC in soil would have derived from root respiration, we would have expected a larger difference in the produced DIC. In fact, during waterlogging, cover crops had a more pronounced effect on measured $CO_2$ fluxes than on DIC, and thus, we think that a significant fraction of the $CO_2$ respired by cover crop roots was released to the atmosphere directly through their stems. Indeed, it has been demonstrated that a substantial amount root-derived $CO_2$ can be transported to the atmosphere via transpiration stream in plant xylem (Aubrey and Teskey, 2021, 2009). The transport of $CO_2$ through Tall
Fescue stems is highly likely, as the species has been shown to develop aerenchyma under waterlogged conditions, enhancing gas exchange between the rhizosphere and the atmosphere (Mui et al., 2021). This feature is specific to Tall Fescue and has not been shown to apply to many other commonly used cover crop species, such as annual ryegrass for example.

If the dissolution of Fe-associated C and root respiration can account only for small portions of the observed increases in TDC, what are the remaining potential sources? In addition to direct release of C from Fe associations, the commonly observed





increase in C solubility upon flooding has been attributed to pH-driven OC desorption from mineral phases (Grybos et al., 2009; Pan et al., 2016), dispersion of soil colloids (Buettner et al., 2014) and accumulation of microbial metabolites and fermentation products in anaerobic conditions (Fairbairn et al., 2023; Tete et al., 2015). The circumneutral initial pH of the soils in our experiment resulted in a slight decrease rather than an increase in pH during waterlogging which was likely caused by an accumulation of $CO_2$ as carbonic acid and bicarbonate ($H_2CO_3$, $HCO_3^-$) (Kronberg et al., 2024). Thus, desorption driven

by the pH increase (Grybos et al., 2009) was unlikely. In soil, fermentation and Fe reduction are often coupled (Lovley 2011; Snoeyenbos-West et al. 2000). Had fermentation proceeded at a much higher rate than Fe reduction in our system, we would have expected a more pronounced initial accumulation of organic metabolites and fermentation products, reflected as an increase in DOC content. Finally, we also want to bring up the possibility that $O_2$ entrapped in soil pores upon water saturation could have formed aerobic microenvironments (Williams and Oostrom, 2000) where aerobic respiration could go-on and

contribute to an increased soil DIC and TDC content. Huang & Hall (2017) speculated that anoxic-oxic interfaces may play an important role in the observed increases in C mineralization during waterlogging periods. The porous and heterogenous soil matrix often leads to significant spatiotemporal variability in soil redox conditions (Fiedler, 2000). Thus, despite the low redox potentials measured in our study (Kronberg et al., 2024), aerobic microsites could have also facilitated aerobic instead of anaerobic respiration during waterlogging.

## 560 4.4 Impact of the cover crop on C mineralization

We expected that Tall Fescue as a cover crop would alter the response of soil $CO_2$ production to waterlogging because of increased substrate availability promoting C mobilization. As opposed to our hypothesis, the response of cumulative $CO_2$ fluxes to waterlogging was similar in both plant treatments. This suggests that cover crop root C inputs did not significantly promote mobilization of OM from mineral phases as reported by e.g. Winkler et al. (2019), although higher dissolved C content

in the cover crop treatment appeared to slightly enhance Fe solubility. The specific feature of Tall Fescue being able to form aerenchyma, which facilitates $O_2$ transport into the roots, may have mitigated its effects on Fe-associated C by accelerating the drop in soil redox potential less than anticipated (Kronberg et al. 2024). Therefore, these findings should not be generalized to other cover crop species.

The applied methodology does not allow us to distinguish soil respiration from total ecosystem respiration which unfortunately

prevents a direct assessment of waterlogging effects on soil versus plant respiration. However, measurements taken during the third waterlogging event, following the removal of aboveground cover crop biomass, showed that cover crop respiration was relatively low (0.6 g C $m^{-2}$ $d^{-1}$) compared to the average difference in $CO_2$ fluxes between monoliths with and without the cover crop (3.2 g C $m^{-2}$ $d^{-1}$). This indicates that the overall higher $CO_2$ emissions from the monoliths with the cover were likely driven by belowground processes rather than by aboveground autotrophic respiration. Furthermore, higher cumulative $CO_2$

emissions with the cover crop persisted when normalized to cumulative above ground biomass further suggesting that the higher fluxes were not resulting from above ground autotrophic respiration. In fact, cover crop biomass was minor compared

to barley biomass (~12%, data not shown) while the root biomass was significantly larger under the cover crop treatment as presented in our previous publication (Kronberg et al., 2024).

Normalizing cumulative $CO_2$ fluxes from the last off-season to root biomass in the top 20 cm of soil showed that the fluxes per gram of root biomass were approximately three times higher without cover crops than with them (Fig. B3). This suggests that although the cover crop increased the $CO_2$ emissions per $m^2$, the plant cover had a net positive effect on soil C sequestration because emissions per soil C input (i.e. root biomass) were smaller. Thus, potential priming effects of the cover crop on OM (not directly measured) did not appear to be substantial enough to offset the positive contributions of the cover crop to soil C storage. Hence, our results tentatively suggest that cover crops had a net positive effect on soil C sequestration, even under periodic waterlogging, which is in line with studies that have promoted cover crops as effective strategy to improve soil C sequestration (Heikkinen et al., 2022; Kaye and Quemada, 2017; Poeplau et al., 2015; Poeplau and Don, 2015).

### 4.5 Environmental implications and research outlook

Our results suggest that temporary waterlogging outside growing seasons might not significantly affect total $CO_2$ emissions or increase $CH_4$ emissions from cultivated mineral soils in cool, humid climates, which is good news from the climate change perspective. A recent study demonstrated that a substantial fraction of agricultural fields in Finland may suffer from periodic waterlogging during late autumns when transpiration is low and rainfall is high (Mattila and Vihanto, 2024). This highlights the urgent need to apply mechanistic insights on the effects of temporary waterlogging and anaerobic conditions on C stability from laboratory incubations to field scale, with our mesocosm study acting as a crucial bridge between the two. However, our study focused merely on the effects of altered soil moisture on C stability and $CO_2$ and $CH_4$ fluxes, without accounting for potential impacts on C inputs, soil N dynamics or on production of another potent greenhouse gas, nitrous oxide ($N_2O$). It is important to recognize that climate change-induced shifts in e.g. off-season soil temperatures and freeze-thaw cycles are also likely to affect greenhouse gas emissions and soil C sequestration capacity (Heikkinen et al., 2022; Liu et al., 2024). Thus, studies that evaluate the net impacts from these factors in an integrated manner are warranted in future.

From the methodological perspective, we want to highlight that the changes in soil $CO_2$ storage during and after high soil moisture events should not be overlooked as neglecting these changes could lead to biased view of momentary soil respiration. In modelling, model-data synthesis is used to match modelled results with the measured fluxes facilitating the calibration of C cycling simulation models (Nevalainen et al., 2022) and enabling the assessment of individual weather parameters' effects on the outcome (Heimsch et al., 2024). The lag between respiration and the release of the respired $CO_2$ to the atmosphere will make such model-data synthesis challenging, as the measured $CO_2$ efflux may result from respiration over the preceding weeks. Incorporating the temporary storage of dissolved $CO_2$ into models would improve the accuracy of the simulation of short-term dynamics and ensure that the respiration terms would be fitted based on the correct time period under when they occur.



In conclusion, our results support previous research suggesting that temporary waterlogging does not suppress $CO_2$ production in mineral soils to the extent predicted by conventional models based solely on aerobic processes. However, as opposed to our hypothesis, the reductive dissolution of Fe oxides and the mobilization of associated C did not alone explain the sustained $CO_2$ production. Furthermore, root C inputs from Tall Fescue, used as an overwintering cover crop, did not promote a substantial release of Fe-associated C, as indicated by low dissolved Fe concentrations and an unaltered soil $CO_2$ efflux response to waterlogging in the presence of plant cover. Overall, our results suggest that off-season waterlogging in cool, humid climate may not significantly affect the annual cumulative C efflux from mineral soils to the atmosphere.

**Author contributions**

**R.K.:** Conceptualization, Formal analysis, Funding acquisition, Investigation, Methodology, Project administration, Visualization, Writing – original draft, Writing – review & editing. **S.K.:** Conceptualization, Methodology, Supervision, Writing – review & editing. **M.P.:** Conceptualization, Methodology, Supervision, Funding acquisition, Project administration, Writing – review & editing. **T.P.:** Methodology, Software. **M.K.:** Methodology, Software, Writing – review & editing. **T.M.:** Supervision, Writing – review & editing.

**Acknowledgements**

This research was funded by Strategic Research Council at the Research Council of Finland (Multi-benefit solutions to climate-smart agriculture, MULTA) (grant number 337549, 328309, 327236), Research Council of Finland post-doctoral researcher funding (grant number 339489), Drainage Foundation sr. (grant number H-13-2022-9.1), and Maa- ja Vesitekniikan Tuki ry (grant number 44875). R.K. acknowledges the University of Helsinki Doctoral Programme in Sustainable Use of Renewable Natural Resources (AGFOREE) for financial support as a doctoral scholarship. We are grateful for Prof. Jussi Heinonsalo (University of Helsinki) for his valuable contributions to the experimental conceptualization and planning. We also want to thank Rauna Lilja and Lisa Leinonen for their contributions to establishing and maintaining the experiment. Figure 1 was created in BioRender. Kronberg, R. (2025) (https://BioRender.com/pg6cizz).

**Competing interests**

There are no competing interests to declare.



**Declaration of Generative AI and AI-assisted technologies in the writing process**

During the preparation of this work R.K. used ChatGPT 4o to improve the readability and language. Sentences were given to ChatGPT after which R.K. reviewed and edited the content as needed. Authors take full responsibility for the publication content.

**Data availability**

The data have been published in Zenodo (https://doi.org/10.5281/zenodo.14438980).



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
