# Peer review of "Temporary waterlogging alters CO2 flux dynamics but not cumulative emissions in cultivated mineral soils"

_EGUsphere, 2025_

## Author Comment (AC2)

Dear Editor and Referees,

Thank you for considering our work for publication in *Biogeosciences*. We highly appreciate the reviewing efforts and the constructive feedback that will help us to improve our manuscript. We have now reviewed the comments and provide thorough responses below. The indicated improvements to the manuscript itself will be made at the next stage of the reviewing process.

We look forward hearing from you soon.

Yours sincerely,

On behalf of the authors
Reija Kronberg and Mari Pihlatie

**Reviewer 1**

The authors present a thorough and well-described study of saturation treatments in soil monoliths (i.e., 3D soil columns) collected from agricultural field sites. The overall manuscript is very well-written, and the description of data analyses is particularly thorough and of high quality: nicely done! However, there are some areas where it is not clear that the focal concepts and hypotheses align with the study design or expectations, leading to a need for some careful consideration of the relevance of cited literature and the justification for hypotheses/interpretations. Specifically, much of the background information and explanation of relevant processes is tied to the idea of Fe-SOC associations driving mobilization of DOC and higher CO2 mineralization. However, there is a key distinction between prolonged saturation and frequent saturation cycling, because the saturation cycling is likely the mechanism for changes in Fe reducibility and subsequent reduction/release of DOC. Further, the importance of this mechanism is contingent on a measurable amount of the SOC in a system being stabilized via Fe associations to begin, and much of the related work has been done in tropical soils or acidic podzols with very high reactive Fe content. In this study, I would characterize the saturation pattern as one relatively short-term but consistently waterlogged within the treatment periods, so the design does not invoke the mechanism of cyclic saturation underpinning many of the related studies (which is acknowledged in the Discussion), and it is not clear whether initial Fe-SOC interactions are a major contribution to SOC stabilization in this system. While I think the overall conclusions and insights from the study are interesting and important, some reframing of the conceptual underpinning is needed to justify the design and interpretations, and some of the known considerations mentioned in the Discussion need to be introduced more directly early on. These points are explained in more detail below, e.g. on comments on lines 25-26, 56-58, and 428-429.

**Author response:** We thank the reviewer for their thorough feedback, which will help us to strengthen our manuscript and to clarify certain key concepts. We acknowledge the importance of distinguishing between prolonged versus fluctuating saturation regimes and will revise the Abstract, Introduction and particularly the Discussion as described in our responses to specific comments. We also recognize the need to better justify the role of Fe−C associations in boreal soils, and will provide better rationale by highlighting the relatively high proportion of shortrangeordered Fe oxides typical of these soils and their demonstrated correlation with mineral-associated organic carbon further highlighting their importance in C sequestration. For more detailed responses please see our responses to Specific comments below.

**Specific Comments**

1. Line 25-26: A key factor related to waterlogging is the cycling of saturated conditions, in addition to the absolute duration. From the abstract alone, this seems like it was repeated waterlogging events over a seven-week period, but it is not completely clear; from the text, it seems that it is continuous waterlogging for a 7 week duration. Suggest more directly describing the timing and duration of waterlogging (i.e., repeated events vs. one continuous 7-week waterlogging condition) here, as it is important context for the subsequent interpretation of results (see later comments, e.g. on lines 56-58).

**Author reply:** We agree that the description of the waterlogging duration requires clarification and will revise the text to explicitly state that the treatment involved continuous waterlogging for a 7-week period during each offseason.

*The 1.5-year study comprised three growth cycles with alternating growing and off-seasons. Spring barley (Hordeum vulgare) was grown in all monoliths during the growing seasons. In turn, during all three off-seasons, half of the monoliths were subjected to waterlogging lasting seven weeks, while in the control monoliths soil moisture was maintained below field capacity.*

2. Lines 26-27: As described, it is unclear whether the same half of monoliths received both overwinter cover crop and waterlogging treatments. As described in the text, these are a factorial design; please clarify here.

**Author reply:** The research design will be clarified in the revised manuscript.

*Within these water treatment groups, the monoliths were further divided into two plant treatment groups: in half of the monoliths, an overwintering cover crop (Festuca arundinacea) was grown, while in the other half the soil was left bare for the off-seasons.*

3. Line 33: "both soils and plant treatments…" Wording is unclear: please revise for clarity. In addition, how the plant treatments overlap with waterlogging treatments is not yet clear, as mentioned above.

**Author reply:** We will revise the manuscript to more clearly describe the setting. In this specific context, the statement regarding the plant treatment is not essential and will therefore be removed for clarity.

*After the initiation of drainage, $CO_2$ fluxes from both soils increased more than predicted based on changes in soil moisture and temperature, likely due to the release of previously accumulated CO2.*

4. Lines 56-58: An important overarching consideration for this work in the context of "temporary" waterlogging is the frequency of saturation cycles, in addition to the total duration of saturation. The process of Fe-associated OM destabilization is described nicely below, but one of the additional mechansims

underpinning some of the observations in the cited studies is the progressive increase in overall Fe reducibility from repeated dissolution/precipitation processes; e.g., as discussed in Ginn et al. (2017) (https://pubs.acs.org/doi/full/10.1021/acs.est.6b05709) and mentioned in the Discussion section. Prolonged vs. fluctuating saturation is one important factor conceptually distinguishing short-term but consistent waterlogging (in this study) from in-situ or manipulated variable moisture conditions, and relevant literature.

**Author reply:** We agree with the reviewer that both the duration and frequency of waterlogging are key factors influencing iron–organic matter interactions under saturated conditions. This important aspect will be introduced earlier in the revised manuscript. However, since our experiment did not specifically address the effects of saturation duration or frequency on carbon dynamics, we prefer not to place too much emphasis on these factors in the Introduction. The same reasoning applies to the discussion of mechanisms that might increase the overall reducibility of Fe under fluctuating redox conditions.

In boreal soils, a large proportion of Fe oxides are poorly crystallized and therefore expected to be easily reducible; therefore, we expect that repeated waterlogging would have a smaller impact on Fe reducibility compared to soils containing more crystalline oxides. In the revised manuscript, we will expand the Discussion to more thoroughly address how saturation patterns may influence iron reduction and Fe–organic matter associations.

5. Lines 90-92: While I agree that many of the more targeted experiments have been conducted in laboratory scale incubation, there have been field studies evaluating fluctuating water conditions related to OM mobilization, e.g. Possinger et al., 2020 (cited in the Discussion). However, field-only observations also have limited ability to have controlled comparisons across treatments, for example. Thus, the value of a middle ground between controlled lab incubations and in-situ studies could be more clearly identified here.

**Author reply:** We appreciate the reviewer's suggestion and agree that revising the text as recommended will help to clarify the rationale behind our research. We will therefore modify the manuscript to more explicitly highlight the value of mesocosm experiments as a bridge between controlled laboratory incubations and observational field studies.

*Thus far, studies focusing on the effects of water saturation/anaerobic conditions on OM mobilization and $CO_2$ fluxes in mineral soils have mainly been either laboratory scale incubations (Bhattacharyya et al., 2018; Fairbairn et al., 2023; Huang et al., 2021, 2020; Huang and Hall, 2017; Zhao et al., 2020) or field studies (Jeanneau et al., 2020; Possinger et al. 2020). The middle ground mesocosm experiments*

*considering the essential role of soil structure (Bergström, 1990; Lehmann et al., 2020) and plant cover (Keiluweit, 2015; Liang, 2023) on C flows and transformations, while maintaining the control over environmental conditions, have been rare. In our 1.5-year greenhouse experiment, we focus on this knowledge-gap by studying C dynamics in intact soil profiles under controlled temperature and water conditions to capture processes more representative of field conditions.*

6. Lines 97-98: An important assumption underpinning the Fe-C hypothesis is that a significant proportion of mineral-stabilized C in this system is actually associated with Fe to begin. Are these soils abundant in redox-active Fe-rich mineralogy? Note that many (not all) of the cited studies were conducted in tropical soils with high Fe oxides (e.g. much of the work by Aaron Thompson et al.), acid forest podzols dominated by Fe- and Al-C complexes, etc. While the more detailed soil description and characterization is more relevant for the methods, the justification for the overall hypotheses and study design/concept is important here, and sets up the subsequent interpretations. See also comment on lines 608-609 in the Discussion on the importance of this assumption and related interpretations.

**Author response:** We acknowledge the importance of addressing the relevance and representativeness of the selected soils in our study, including comparison of the studied northern soils to other studied soils mostly representing tropical soils or podzols rich in Fe- and Al-C complexes. We will revise the introduction to provide a stronger rationale for studying the impact of waterlogging on Fe–C associations in our northern agricultural soils. Specifically, we will highlight the relatively high abundance of metastable short-range-ordered Fe oxides in boreal soils and the reported correlation between Fe oxides and soil C content. In addition, we will clarify that the soils chosen for this study are representative of cultivated boreal soils in the region.

Here we summarize insights to the importance of the study, part of which will also be integrated in the revised version. First, boreal soils typically contain a relatively large proportion of metastable, low-crystallinity Fe oxides compared to their total Fe oxide pool. While it is true that the soil types used in our study are not especially rich in Fe minerals, particularly if compared to tropical or volcanic soils frequently employed in incubation studies, the young age of boreal soils (Donner, 1995) results in a substantial fraction of short-range-ordered oxides (SRO) oxides (Niskanen, 1989; Peltovuori et al., 2002). These oxides are more easily reducible than their crystalline counterparts and thus especially important for C stabilization and release processes.

The two soils included in our experiment represent typical organic matter (Fernandez-Ugalde et al., 2022; Heikkinen et al., 2013; Lemola et al., 2018) and SRO Fe oxide contents for the region (Niskanen, 1989; Peltovuori et al., 2002). SRO oxides (measured by acid ammonium oxalate extraction) accounted for nearly half of all pedogenic Fe (measured by citrate-bicarbonate-dithionite extraction, see Kornberg et al. (2024)). This proportion is comparable to values reported in Andosol rich in

metastable Fe phases in Winkler et al. (2018, 2019) who reported SRO/total pedogenic Fe ratios of <0.4. The SRO Fe concentration in Andosol was 17 g kg$^{-1}$. By comparison, our soils contained approximately 6 and 3 g kg$^{-1}$ SRO Fe oxides in silty clay and sandy loam soils, respectively which is lower, but of the same order of magnitude.

Second, despite representing a minority of the clay-sized fraction in humid boreal soils, SRO Fe and Al oxides play a disproportionately important role in C stabilization due to their high specific surface area and chemical reactivity (Kaiser and Guggenberger, 2003; Yong et al., 1992). A recent large-scale study of arable soils in southern Finland (97 fields) showed that **Fe oxides correlated with mineral-associated organic carbon**, particularly in soils with high clay contents (>30%; Salonen et al., 2024), such as our silty clay soil.

Taken together, these factors suggest that even in soils with modest total concentrations of Fe oxides, their high reactivity and substantial proportion in poorly crystalline forms make them important reactive components in boreal agroecosystems. At a time when C availability and mineral associations are increasingly recognized as key determinants of soil C turnover, our study advances understanding of C–mineral interactions and their implications for C cycling under changing climatic conditions. Results on Fe-associated C content measured after the three growth cycles in this study will be presented in a subsequent publication focusing specifically on the distribution of final C fractions.

7. Lines 112-113: More detail and justification for soil hypotheses related to Fe-C stabilization is needed. (1) In my opinion, "tentative" classification is insufficient: how was this assigned (by soil maps)? Do these soils align with characteristics of the assigned soil types? (2) Building on the comment regarding soil Fe abundance and Fe-OM associations, neither of these orders are archetypically associated with especially high Fe abundance or Fe-dominated OM stabilization to my knowledge. This isn't to say that they do not have abundant reactive Fe, but providing the reader some context for the oxalate-extractable Fe and Al presented in Table 1 would help to justify that there is significant contribution of these phases to OM stabilization in this system. (3) Is there a fundamental difference in the saturation regime between the Stagnosol and Cambisol/Umbrisol? In addition to texture, this seems a key factor that could influence interpretation of subsequent response to waterlogging and should be clarified here.

**Author response:**

1) The soils were classified according to the World Reference Base for Soil Resources (IUSS Working Group WRB, 2014) system, endorsed by EU. The word *tentative* was used due to limitations in available soil data used in soil classification (see Kronberg et al. (2024)). More specifically, the

height/depth of the soil profiles used in soil classification were only 80 cm instead of 1 m deep. Thus, the deeper horizons, also relevant for classification, were not available. Therefore, particularly the possible signs of high groundwater table (gleyic color pattern) could not be explicitly observed. Additionally, possibilities in recording the structure were limited. However, these shortcomings do not influence the classification at the highest level. The classification procedure is described in Kronberg et al. 2024 (Supplementary information).

The word *tentative* will therefore be removed to avoid confusion.

2) This point is linked to the previous comment regarding lines 97–98. As noted in our response to Comment 6, we will revise the introduction to provide improved contextualization. We will emphasize the relative abundance of short-range-ordered Fe oxides in boreal soils, the observed correlation between Fe oxides and soil C content, and the fact that the studied soils are representative of cultivated boreal soils in Southern Finland.

3) Indeed, the two soil profiles differed in their hydrological characteristics. The dense structure and fine texture in the silty clay soil limited the movement of water through the profile causing water stagnation and repeated alteration of oxidized and reduced conditions resulting in the formation of stagnic properties (see Kronberg et al. 2024). In contrast, the coarse texture in the sandy loam soil facilitated an efficient water movement and transport of solutes into deeper soil layers. In the profile brown colors indicated a release of Fe from primary minerals.

We will add this information into the manuscript and briefly consider it in the discussion.

8. Lines 121-122: This is a minor point and mostly a difference in usage/terminology, but a monolith in my understanding is a 2D representation of a profile surface, rather than a 3D cylindrical core. I would interpret the soil systems as described as soil columns or mesocosms. I would suggest describing once early on in the text that you're referring to columns; e.g. "soil monoliths (i.e., 3D soil columns)..." or similar.

**Author response:** We must respectfully disagree with the reviewer on the usage of terminology. In several studies the term *monolith* has been used to specifically refer to undisturbed soil profiles/columns as opposed to repacked soil columns (e.g. Bergström, 1990; Herbrich et al., 2017; Lewis & Sjöström, 2010; Virtanen et al., 2013). However, we agree that the terms (profiles/monoliths) could be clarified especially in the abstract. We will

substitute the term *profile* with *monolithic soil columns* that will thereafter be referred to as *monoliths*.

9. Lines 162-164: How was waterlogging maintained after full saturation was reached throughout the 54 and 50 day periods, especially between cover crop and no cover crop treatments? If the columns were watered subsequently, did partial drying occur during watering events?

**Author response:** Waterlogging was maintained by manually watering the monoliths based on soil moisture readings from Teros12 sensors. Despite careful monitoring, slight oscillations in soil moisture occurred due to cover crop transpiration, particularly in the sandy loam topsoil. This introduced a minor confounding effect on soil redox state, as noted in our preceding study (Kronberg et al., 2024). While we consider the impact on C fluxes to be small, it may have facilitated more efficient gas transport from soil to the atmosphere. We appreciate the reviewer's observation and will revise lines 565–568 to include this potential mechanism when discussing the influence of cover crops on soil C dynamics.

10. Line 214: With high TDC solutions, freezing can sometimes induce coagulation/flocculation of organic solids which do not always resuspend following thaw. In addition, I am not familiar with the efficacy of freezing alone for DIC, which is sensitive to exposure to atmosphere. Could you comment a bit further on the preservation approach and potential limitations, if any, for TDC/DOC/DIC analyses?

**Author response:** We were aware that freezing could cause coagulation/flocculation of dissolved organic carbon, but we did not observe any signs in our samples. Overall, freezing has been reported to have a larger impact on the quality than the total quantity of dissolved organic matter (Chow et al., 2022). Acidification ($HCl/H_2SO_4$) of samples to pH < 2 would have been another preservation approach. However, this, in turn, may have led to organic matter hydrolysis and thus, a loss of dissolved organic matter (Chow et al., 2022). Also, as we were additionally interested in dissolved inorganic carbon, this approach would not have been feasible. In near neutral pH, large fraction of the dissolved $CO_2$ was present as $HCO_3^-$ and thus, sample acidification would have reduced DIC concentration by transforming it to $CO_2$ and leading to more efficient degassing.

Freezing and overall sample handling likely slightly lowered the absolute DIC concentrations because the $CO_2$ concentration in the soil solution slowly equilibrates with the atmosphere. However, we were mainly interested in temporal trends and the differences between different treatment levels than on absolute concentrations. Thus, because the effect of sample storage on DOC/DIC concentrations was assumed to be

uniform across the samples, the overall impact on our observed trends and conclusions is considered minimal.

11. Line 215: The data analysis and statistical approaches are very thorough. Nicely done! In some cases, they could be slightly trimmed down for readability when discussing conversions (if standard to the field), e.g. in the section on DIC change calculations.

**Author response:** This section will be revised and compressed as recommended by the reviewer.

12. Lines 229-230: Can you expand a little more on this interpolation? What is the linear part of the interpolation based on (between two sampling dates)? I can guess at how na.approx might work for this, but it's not immediately clear how this would be done with a sequence of measurements over time. It's not a major point, but a little clarification would be helpful for replicability of the method.

**Author response:** The sentence will be revised to improve clarity.

*To calculate cumulative $CO_2$ fluxes, we first generated daily flux values for each monolith by linearly interpolating between temporally consecutive measurement events using the na.approx function in the zoo package.*

13. Lines 239-241: A sentence before explaining the overall approach for using a modeled $CO_2$ efflux would help provide the reader some context (e.g., to compare measured and modeled efflux to test XYZ...).

**Author response:** We will modify the sentence as suggested by the reviewer

*We tested how well could a commonly used simple empirical model predict the $CO_2$ efflux during and after waterlogging events.*

14. Line 432-434: Figure 2 caption: While the modeling procedure was explained in the text, it would be helpful to remind the reader what "monitored monoliths" means here - it's not immediately intuitive and adds a degree of confusion that may not be needed. In addition, the importance of the difference between Cycle 2 and Cycle 3 is highlighted by having them in two separate panels, but how these experimentally-induced cycles relate to important experimental conditions/phases (e.g., the removal of biomass in Cycle 3) isn't really clear in this figure or how the results are presented in the text. Some additional labeling/timeline explanation in the figure would be helpful to follow the results as presented in the text.

**Author response:** We appreciate the reviewer's feedback. To avoid unnecessary confusion, we will remove the reference to the monitored monoliths from the figure caption. To give readers a clearer understanding of the temporal sequence of cycles

and waterlogging events, we will also add the corresponding data collection period (dates) to each figure title (e.g., *Cycle 2: xx.xx.2021*). Additionally, the figure caption will be clarified to state that the x-axis represents the number of days since the onset of the respective waterlogging event.

In response to another reviewer's comment, we will further revise the figure by adding a panel showing soil moisture in the top 20 cm layer for the same period. Consequently, each soil type will be presented in its own sub-figure.

15. Lines 380-382: It is interesting to see the change in DIC accounting for the main change in TDC. Given the question about storage and sample preservation mentioned above, just as a check: do these values align with typical ranges of DIC in these soil types (especially the values close to or slightly higher initially than DOC in the silty clay soil)?

**Author response:** The elevated DIC concentrations in silty clay soil likely result from the soil's fine texture and dense structure, particularly in the subsoil, which limits gas diffusivity. Consequently, $CO_2$ produced by soil respiration is not released to the atmosphere as efficiently as in the better-aerated sandy loam. This reduced gas exchange promotes $CO_2$ accumulation in the soil solution, thereby maintaining higher DIC concentrations ($CO_2 + HCO_3^-$).

Previous studies have also reported a relationship between soil texture and DIC concentration, with finer-textured soils generally exhibiting higher DIC levels. This pattern has been attributed to longer water residence times in such soils (Rantakari et al., 2010).

Overall, DIC is less frequently measured in boreal soils and catchments compared to DOC, as DOC typically represents the dominant fraction of dissolved carbon in discharge waters from forest and peat soils with low pH (Räike et al., 2016). However, in studies that have included DIC measurements (Räike et al., 2016), concentrations in croplands have been found to be elevated, particularly during low-flow periods, which aligns with our observations.

16. Lines 428-429: Following my comments on the introduction, it's great to introduce the distinction between prolonged vs. fluctuating anaerobiosis here, but I would suggest accounting for this distinction earlier in the text with respect to relevant literature and expectations for the experiment, as it did not apply fluctuating saturation and the related implications for increasing Fe reducibility are less applicable (e.g., see Ginn et al., 2017: https://pubs.acs.org/doi/full/10.1021/acs.est.6b05709).

**Author response:** We thank the reviewer for emphasizing the importance of considering the potential effects of static versus fluctuating redox conditions on C

dynamics. We will revise the manuscript to briefly introduce the mechanisms underlying the processes behind the observed differences between e two circumstances and discuss our results in this broader context.

Ginn et al. (2017) reported that redox fluctuations can enhance Fe dissolution rates by increasing Fe reducibility during successive reduction–oxidation cycles. They suggested that when redox fluctuations occur on similar timescales to ongoing soil processes, they may facilitate greater Fe dissolution and, consequently, the release of associated C.

However, as noted by Winkler et al. (2018, 2019), repeated redox fluctuations may also lead to the depletion of labile C sources. This depletion could, in turn, hinder Fe reduction and thereby confound the extent of reductive Fe dissolution.

As discussed in our response to Comment 6, the studied soils contained a high fraction of low-crystallinity, short-range ordered (SRO) Fe oxides. Therefore, we expect that these SRO Fe oxides could serve as electron acceptors, supporting reductive Fe dissolution and the potential release of associated C even under prolonged anaerobic conditions, particularly in the silty clay soil, which contained a higher abundance of Fe oxides. Additionally, cover crop root litter and exudates likely provided labile substrates that further promote these reductive reactions.

Overall, we will revise the manuscript to ensure that these aspects are clearly presented and discussed in relation to our findings. However, we will do this with more detail only in the Discussion instead of introduction section. As also earlier stated by the reviewer, our experiment did not contain different saturation duration or frequency treatments. We therefore think that setting emphasis on the discussed mechanisms would bring in unnecessary layer of complexity in the Introduction.

17. Lines 433-434: Given the dependence of anaerobic response on soil type as mentioned above, initial characteristics of the soils are critical for interpretation of differences (or lack thereof). For example, though these soils do differ in Fe oxide content, it is not clear from the presented soil information whether Fe oxides in general are a major contributor to overall soil mineralogy in either soil, and further if they contribute meaningfully to SOC stabilization in this system. While the absolute values for the extractable Fe oxides are presented and the reader could convert the mmol per kg values and look up ranges for other soil types, it would be helpful for the reader to have it stated upfront (before or as part of the hypotheses) that these soils are appropriate for testing the hypothesized response due to the contribution of Fe oxides to SOC in this system.

**Author response:** Please see our responses to Comments 6 and 7.

18. Lines 463-465: Here and throughout, it would be helpful to have a little more guidance in the text about **what the reader should take away from statements about observations in a certain study cycle**. The meaning of the reference to the second and third study cycle is not immediately understandable in terms of what that means for the conditions of the experiment; in other words, what are the critical differences between these phases relevant to the target processes? It seems to me that in the third cycle the monoliths have already experienced waterlogging in the first and second cycles, so that is one important difference, as is the removal of biomass, but it's a little bit challenging as a reader to follow these processes.

**Author response:** We appreciate the reviewer's comment and acknowledge that the rationale behind repeating the same treatment several times is not clearly stated in the current manuscript. The rationale was twofold: (a) to increase the robustness of our findings, and (b) to examine whether repetition influences the outcomes, particularly in the monoliths with cover crops, where observable effects on soil properties may require more time to develop.

To address this, we will revise the last paragraph in the Introduction to provide readers with a clearer understanding of this experimental design choice. We will also revise the *Discussion* section accordingly to ensure that it links to the modifications made in the *Introduction*, highlights the key differences observed (or the absence thereof), and clarifies the implications of these outcomes.

19. Lines 608-610: In line with comments above, this take-away is stronger if the experiment were targeting Fe-associated SOC stabilization and dissolution: i.e., with relatively high contribution of Fe-associated SOC (which is unclear from the data presented) and fluctuating conditions that result in subsequently higher Fe reducibility. Consequently, some nuance regarding this take-away might be needed.

**Author response:** Both reviewers agreed that the final conclusions could be slightly refined. We have now emphasized the role of reduced diffusion in the outcome.

*We conclude that because surface $CO_2$ fluxes reflect not only production but also temporary accumulation or release of $CO_2$ within the soil profile, simple empirical models are unable to capture momentary soil respiration. Moreover, the continued $CO_2$ production observed under anaerobic, water-saturated conditions supports previous findings that temporary waterlogging does not suppress $CO_2$ production in mineral soils to the extent predicted by models that rely solely on aerobic processes. However, despite the important role of short-range-ordered Fe-oxides in C stabilization in the studied soils, their dissolution and the mobilization of associated C did not alone explain the sustained $CO_2$ production. Furthermore, root C inputs from Tall Fescue, used as an overwintering cover crop, did not promote a substantial*

*release of Fe-associated C, as indicated by low dissolved Fe concentrations and an unaltered soil $CO_2$ efflux response to waterlogging in the presence of plant cover. Overall, our results suggest that off-season waterlogging in cool, humid climate may not significantly affect the annual cumulative C efflux from mineral soils to the atmosphere.*

Minor Editorial Suggestions:

20. Line 198-199: The detail of CO2 flux conversion can be omitted, unless there was a non-standard conversion used.
21. Line 275: "similarly than with..." Awkward wording here, suggest rewording for clarity.
22. Lines 288-289: Suggest clarifying that this refers to "higher than the non-saturated treatment."

**Author response:** We will perform the suggested revisions a suggested by the reviewer.

**Reviewer 2**

The manuscript submitted to *Biogeosciences* presents an interesting experiment aimed at investigating the effects of temporary waterlogging on CO2 (and CH4) production and release from mineral soils. To achieve this, waterlogging episodes were simulated in the laboratory using soil cores from two sites with contrasting soil textures (silty clay and sandy loam). The authors argue that CO2 production is sustained during waterlogging, while enhanced CO2 release during drainage suggests that diffusion plays a key role in regulating the emission of CO2 previously accumulated in the soil profile. Overall, the study concludes that waterlogging did not significantly alter cumulative CO2 fluxes.

The manuscript is well-written, with a clear structure and easy-to-follow narrative. However, I have identified several general aspects that could be improved, which I outline below. Additionally, I have included specific comments further down

**General comments:**

I recommend simplifying the hypotheses and focusing on testable statements. Explanations for why specific behaviors are expected belong in the introduction and/or discussion. If these explanations are not directly tested, they should not be part of the hypotheses.

I recommend the authors to reconsider what is the role of the CH4 fluxes for your story. CH4 is mentioned in the abstract, and one hypothesis addresses it, but the manuscript provides almost no results or discussion on the effects of waterlogging on

CH4 fluxes. I recommend either dropping CH4 entirely or fully integrating it into the study. The latter would require introducing the topic in the introduction, presenting results, and discussing potential mechanisms behind the observed patterns.

The results section is currently too lengthy and includes elements of discussion. I suggest moving discussion points to the appropriate section and succinctly describing only the results relevant to the study's argument. Additionally, I had difficulty locating some referenced tables and figures (e.g., Fig. B1, Table A3). Please ensure all references to figures and tables are accurate and consistent.

Finally, I consider that several treatments are not significantly different between them due to the large heterogeneity in the fluxes and within the replicates, but there is room for a detailed interpretation of these trends, while keeping caution in the formulation. In Figure 6 (cumulative fluxes) I do see a trend towards: 1) higher $CO_2$ emissions in <FC compared to saturation; 2) higher $CO_2$ emissions in the silty clay compared to the sandy loam and 3) higher emissions in CC (for the third waterlogging).

**Author reply**: We thank the reviewer for careful evaluation and constructive feedback, which will help us improve the manuscript. We agree that the hypotheses should be simplified and we thus present modified versions in this response letter. Additionally, as methane currently plays a minor role in the manuscript, we have decided to leave it out of the hypotheses.

Regarding the reviewer's concern that the Results section is too lengthy, we will revise and condense Section 3.2 ($CO_2$ efflux modelled with the empirical model) to align with the corresponding modifications planned for the Discussion section. Overall, however, we consider the majority of the results currently presented to be essential for understanding the issues discussed later in the manuscript and for providing the necessary context for interpreting our findings. After implementing the suggested modifications, we will re-evaluate the section to determine whether additional compression or removal of content is warranted.

Overall, many of the issues raised by the reviewer in the General comments are also addressed in the Specific comments, where we provide detailed responses to each point.

**Specific Comments:**

i. **L105**: Hypothesis (b) includes two separate hypotheses (one about $CO_2$ and one about CH4). The first part is overly complex and involves causation mechanisms that cannot be tested with the current setup. I suggest simplifying the hypothesis to focus on what can be tested. For the CH4 component, ensure the topic is introduced in the introduction (see general comments on CH4).

**Author response:** We will revise the hypotheses according to reviewers recommendation and focus on statements that can be tested with our experimental setup. The hypothesis regarding the methane production will be dropped out.

Revised hypotheses:

a) *Temporary waterlogging mobilizes Fe-associated C which leads to increased soil DOC content.*

b) *Temporary waterlogging does not reduce cumulative $CO_2$ emissions from either soil.*

c) *Higher OM, clay and Fe oxide contents result in higher cumulative $CO_2$ fluxes from silty clay than from sandy loam soil.*

d) *Soil C input from cover crops promotes C dissolution and $CO_2$ production in waterlogged conditions.*

    ii. **L158**: Please clarify which depth was used as a reference.

**Author response:** We are not entirely sure what the reviewer means here. We will clarify that we aimed to maintain soil moisture at 50% WFPS in the topsoil (top 20 cm). In deeper layers moisture remained higher in both soils (see Kronberg et al. 2024).

    iii. **L162**: This section is unclear. What exactly is meant by "events"? How long did they last? While it is stated that all monoliths were irrigated with up to 23 mm, it is not entirely clear if this was consistently applied.

**Author response:** We agree with the reviewer that this statement was not clear, and we will therefore clarify it in the revised manuscript. Basically, monoliths were irrigated with 0–23 mm water/monolith per manual irrigation, and this was repeated several times per day if needed. Irrigation was stopped/paused once water infiltration ceased. Soil in silty clay monoliths got saturated already during the simulated heavy rainfall whereas in sandy loam saturation took on average eight days longer.

*After, these monoliths were manually irrigated with 0–23 mm water/monolith/irrigation, and this was repeated several times per day if needed. Irrigation was stopped/paused once infiltration ceased. Soil in silty clay monoliths got saturated already during the simulated heavy rainfall whereas in sandy loam saturation took on average eight days longer.*

    iv. **L174**: Replace "speeds" with "flow rates."

**Author response:** Will be replaced as suggested.

    v. **L175**: This information does not align with L169. Given that the cross-sectional area is constant and the height varies by a factor of 2, why does the volume vary by a factor of 3.5? Please clarify.

**Author response:** We thank the reviewer for pointing out this inconsistency. The given volumes include the monolith headspace. We will correct the values to match the chamber volumes.

    vi. **L180**: How were leakages observed or identified?

**Author response:** Leakages were evident in the soil moisture data and could also be identified by observing the sensor/porewater collector seams on monolith walls.

    vii. **L190**: The term "visual inspection" is misleading here, as this describes a numerical comparison.

**Author response:** We will remove the first part of the sentence.

    viii. **L192**: This sentence is unclear. You mention discarding data with R2< 0.84 (0.6% of the data). If this threshold was applied, the fluxes below it would be discarded regardless of the reason. Please clarify the purpose of this analysis.

**Author response:** We will replace the current version of this procedure with a revised one described below.

Linear fits of $CO_2$ mixing ratios against elapsed time were initially examined visually. Clear deviations from linearity were interpreted as evidence of leakage, typically caused by improper chamber insertion. Erroneous measurements were excluded by applying an R2 threshold of 0.84. This threshold effectively removed all failed closures in cases where $CO_2$ flux exceeded three times the standard deviation of the flux measured in the empty chamber. For lower fluxes, R2 values were generally below 0.84 even when the measurement was successful. These instances were therefore evaluated individually and excluded only when the $CO_2$ mixing ratio exhibited evident fluctuations indicative of leakage (0.4% of the total data).

    ix. **L200**: What is the purpose of differentiating aboveground and belowground CO2 fluxes in the context of this study? Additional context would be helpful.

**Author response:** We will revise the section as described below to include the rationale for this determination.

As only dark chamber measurements were conducted, total ecosystem respiration, including $CO_2$ originating from both soil and plant components, was obtained. To estimate the fraction of $CO_2$ derived specifically from the soil, aboveground cover crop biomass was removed 43 days after waterlogging during the third study cycle. $CO_2$ fluxes were measured twice on the same day, immediately before and after biomass removal. The difference between the two measurements was assumed to represent respiration from aboveground plant tissues. This approach was applied to ensure that the observed differences between the cover crop and no cover crop treatments did not derive simply from the variation in aboveground biomass.

x. **L204**: What is the temporal resolution of these measurements? This information is missing.

**Author response:** The measurement was repeated right after the biomass removal. This information will be integrated in the revised text.

xi. **L227**: It is unusual to inspect assumptions visually when established statistical tests are available. Consider using these tests instead.

**Author response:** Here we respectfully disagree with the reviewer. Visual inspection is generally considered a viable method for assessing normality and homoscedasticity (e.g. Osborne & Waters, 2002), and indeed it is sometimes a better method than mechanistic testing (e.g. Shatz, 2024).

xii. **L240**: Four references seem excessive here. Consider reducing them.

**Author response:** We will reduce the references to two (Luo and Zhou, 2006; Moyano et al., 2013).

xiii. **L291/Figure 2**: Why were the results for water content not presented? I suggest integrating them into Figure 2. Additionally, fine-tune the figure by removing the area with negative time values.

**Author response:** The soil water content data have been presented in our preceding publication (Kronberg et al., 2024). However, we agree that incorporating soil moisture dynamics into the $CO_2$ flux figure would enhance the interpretation of the relationship between these variables. We will therefore revise the figure to include a panel displaying soil moisture below the $CO_2$ flux plot. In the revised figure, data will be organized by soil type and experimental cycle. Within the faceted layout, plant treatment and variable will be presented in columns and rows, respectively. This design will yield four subfigures instead of the current two.

xiv. **L331**: Remove the word "statistically."

**Author response:** Will be removed as suggested.

xv. **L336, Table 2**: Do you have a statistical test to show whether differences in $Q_{10}$ values between cover crop and no cover crop are significant?

**Author response:** We did not perform statistical tests to evaluate differences in $Q_{10}$ values between seasons. These differences could be assessed using the 95% confidence intervals, which show slight overlap among treatments. Wald's test could also be applied to compare model fits. However, based on other reviewers' suggestions, we will revise the manuscript by removing the standalone paragraph on model performance and integrating its key points into the section on the effects of waterlogging on $CO_2$ production and efflux. To improve coherence, we will also remove the section discussing $Q_{10}$ values, as it falls outside the main scope of our

study. We believe these changes will strengthen the manuscript's central message and streamline the overall Discussion.

> xvi. **L342, 344**: What is Fig. B1?
>
> xvii. **L352, 355**: What is Table A3?

**Author response:** We thank the reviewer for pointing out the false references to supplementary tables and figures. Initially, the supplementary tables and figures were organized in Appendix A and Appendix B. However, during submission, these materials were combined into a single supplementary file. The previous appendix references were inadvertently left in the manuscript. This will be corrected in the revised version. Currently, Fig. B1 refers to Figure S1 and Table A3 to Table S3.

> xviii. **L415/Figure 6**: While the authors claim no significant differences between FC and saturation, there is a clear trend toward higher emissions in <FC, which should be acknowledged. Additionally, why are there one and two asterisks in the same panel (upper left-hand panel)? Finally, I suggest avoiding the term "respired fluxes" for cumulative fluxes, as this could be misleading.

**Author response:** The difference between FC and saturation treatments is already acknowledged in the preceding paragraph. In Figure 6, results from the pairwise comparisons of the means in the two water treatments are presented. The difference was statistically significant only in Silty clay during study cycle 2. We agree that the presentation of the statistical results may be misleading, and we will therefore modify the graph by replacing the current asterisks with only one in between the two bars. The respired CO2-C will be changed to produced $CO_2$-C.

> xix. **L425**: This sentence and its references are confusing. The authors state that their results align with several studies (which ones?) but contradict conventional model assumptions (which ones?). Separate the references to clarify who found what.

**Author response:** We will remove the latter part of the sentence to improve clarity.

> xx. **L433**: See general comments. There is a trend suggesting a soil type effect that should be discussed.

**Author response:** We respectfully point out that the soil type effect was only observed in the $CO_2$ production during waterlogging (Figure 6) but not in total cumulative $CO_2$ fluxes calculated for the waterlogging and the following drainage period (Figure 3). It should also be noted that higher $CO_2$ production in the silty clay soil was restricted to the second study cycle. Since this slight effect was limited to a single study cycle and was not reflected in the cumulative fluxes over the entire inspection period, we prefer not to overemphasize this minor and inconsistent trend to maintain focus and conciseness in the manuscript. However, we agree that the

observed soil type effect on $CO_2$ production during the waterlogging phase merits a mention. We will therefore add a clarifying sentence in the revised version to acknowledge the observation, as follows:

*Contrary to our hypothesis, the cumulative $CO_2$ fluxes (calculated over the waterlogging and the following drainage period), and their response to waterlogging, were similar in the two soils. A slight difference between the two soils was only observed in $CO2$ production during waterlogging in the second study cycle. The lack of clear differences between the two soils contradicts with the higher DOC and DIC contents measured in the silty clay than in the sandy loam topsoil during waterlogging implying a higher mineralization rate in the silty clay. The dissolved C dynamics in deeper soil layers, however, help to explain the lack of difference in the aggregated $CO_2$ fluxes from the whole profile.*

xxi. **L452**: Either remove "in determining" or add "considering" before "the following drying period." Avoid using "soil C budget" in this context; instead, use "soil CO2 efflux" or a similar term.

**Author response:** We will modify the sentence as suggested.

*Thus, our results underscore the importance of considering the following drying period when assessing the impacts of waterlogging on soil $CO_2$ efflux under fluctuating moisture conditions.*

xxii. **L454**: This paragraph relies heavily on a previous publication, and the lack of detail in the CH44 results suggests that it might be better to drop CH4 from the manuscript entirely.

**Author response:** We appreciate the reviewer's suggestion and agree that the hypothesis related to CH4 production can be removed. Accordingly, we will revise this section to reflect that CH4 is not specifically investigated in the revised manuscript.

Previous incubation studies have demonstrated that methane production can contribute substantially to soil C mineralization under anaerobic conditions in mineral soils (e.g. Huang et al., 2021). For this reason, we consider it appropriate to briefly acknowledge this process. In the revised manuscript, the main message of the paragraph will be that methanogenesis under waterlogged conditions can compensate for reduced $CO_2$ emissions, influencing overall carbon mineralization but that in our study no methane fluxes were detected.

xxiii. **L462-480**: I would welcome a discussion on why the models work or fail and how this could be addressed. This likely relates to the integration of soil moisture in the models. However, instead of discussing water content changes, the focus is on Q10, and the paragraph concludes that the experimental design is not appropriate for this purpose. The take-home message is unclear and should be streamlined.

**Author response:** We agree with the referee that the current discussion on model performance is not fully aligned with the scope of our study and that the take-home message was insufficiently clear. Accordingly, we will revise the manuscript by removing the standalone paragraph on model performance and integrating its key points into the section discussing the effects of waterlogging on $CO_2$ production and efflux. This restructuring will strengthen the take-home message, improve coherence, and streamline the overall Discussion.

4.3.1 Waterlogging causes a discrepancy between produced $CO_2$ and measured flux

*The response of $CO_2$ efflux to soil moisture reflects not only the moisture sensitivity of heterotrophic respiration but also the associated effects on gas transport. When soil pores become filled with water, diffusive gas movement is substantially impeded (Greenway et al., 2006). Accordingly, respired $CO_2$ becomes trapped within water-filled soil pores, leading to a discrepancy between $CO_2$ production and measured soil $CO_2$ efflux (Jeanneau et al., 2020; Maier et al., 2011, 2010; Sánchez-Cañete et al., 2018). In our study, increasing soil DIC concentrations—reflecting the amount of $CO_2$ retained within soil pores — indicating that during waterlogging, C mineralization continued at a higher rate than would have been inferred from $CO_2$ efflux measurements alone. These observations demonstrate that, contrary to convention, momentary soil $CO_2$ efflux should not be considered equivalent to soil respiration as the two are decoupled in periodically high soil moisture events (Maier et al., 2011; Ryan and Law, 2005; Sánchez-Cañete et al., 2018). In the long term, the two can usually be considered equal as all respired $CO_2$ will eventually be released into the atmosphere (Maier et al., 2011). However, momentary fluxes can significantly deviate from respired $CO_2$ (Bond-Lamberty et al., 2024; Maier et al., 2011, 2010; Sánchez-Cañete et al., 2018). In our experiment, accumulated $CO_2$ ($\Delta DIC$) during waterlogging was at most ~30% of the total $CO_2$ production (sum of cumulative $CO_2$ fluxes and $\Delta DIC$), being in a good agreement with previously reported values during heavy rainfall events (Maier et al., 2011; Sánchez-Cañete et al., 2018).*

*The discrepancy between $CO_2$ production and $CO_2$ efflux measured at the soil surface caused the empirical $CO_2$ efflux model to fail in simulating the flux dynamics and capturing the post-drainage $CO_2$ pulse.* As a result, the model was unable to reproduce the $CO_2$ pulse upon drainage because the pulse did not represent momentary soil respiration but was largely a consequence of the release of previously respired $CO_2$. *Thus, the observed decrease in $CO_2$ efflux likely stems from a decreased gas transport rather than production only. These results support previous evidence that simple models may underestimate the $CO_2$ production during water saturated, anaerobic conditions (pitäisikö olla viite?).*

*To account for the difference in measured $CO_2$ efflux and soil respiration, Maier et al. (2011, 2010) and Hirsch et al. (2004) have proposed an incorporation of a storage flux term in soil moisture dependency models. They defined the storage flux as the*

*flux resulting from changes in the amount of CO₂ stored in soil (Hirsch et al., 2004; Maier et al., 2011, 2010). We believe that incorporating this storage flux term, could have also improved model performance in our study. However, a robust implementation would have required higher spatial resolution of DIC measurements across the soil profile. Overall, to accurately model C dynamics during transient or temporary waterlogging, process-based approaches that account for anaerobic CO₂ production (Fairbairn et al., 2023) and changes in gas transport should be considered.*

xxiv. **L463**: The term "conventional CO2 efflux model" is mentioned several times. Please define it clearly.

**Author response:** We will clarify this in the revised manuscript.

xxv. **L488**: Highlight the important idea that there is a transport process between CO2 production (respiration) and CO2 release (efflux).

**Author response:** Please see our response to comment xxiii.

xxvi. **L507**: This section overlaps with L462-480. Consider streamlining the discussion to avoid redundancy.

**Author response:** Please see our response to comment xxiii. The structure of this section will be improved in the revised manuscript.

xxvii. **L539**: Was this process captured by your experimental setup?

**Author response:** This process was not directly tested in our experiment. In the revised manuscript, we will clarify that our experimental design does not allow us to demonstrate that aerenchyma in tall fescue facilitated CO₂ transport to the atmosphere but rather provides a plausible mechanisms that helps to explain our observations.

**L579**: Were these results presented in the results section? If not, they should be included.

**Author response:** The results of the cumulative CO₂ fluxes from the last off-season normalized per root biomass are currently presented in Figure S3 and used in the Discussion section. In the revised manuscript, we will represent these results already in the Results section together with other results regarding the cumulative CO₂ fluxes.

xxviii. **L595**: The study does not appear to address C stability. Instead, it focuses on DIC and DOC, which are more related to C accessibility.

**Author response:** We appreciate this clarification. We will replace the term *stability* with *dynamics*, as it more accurately reflects the results presented in this study.

Findings related specifically to C stability will be reported in a forthcoming publication.

> xxix.    **L608-613**: The mismatch between modelled and measured $CO_2$ fluxes likely stems from the neglect of diffusion processes rather than the sole reliance on aerobic processes. Regarding the effect of cover crops, see my earlier comments, as I suspect some marginal differences are being overlooked.

**Author response:** We agree that both factors likely contribute to the observed mismatch. Because surface $CO_2$ fluxes reflect not only production but also temporary accumulation or release of $CO_2$ within the soil profile, simple empirical models are unable to capture momentary soil respiration. Moreover, the continued $CO_2$ production observed under anaerobic, water-saturated conditions supports previous findings that temporary waterlogging does not suppress $CO_2$ production in mineral soils to the extent predicted by models that rely solely on aerobic processes. We will clarify the Conclusions section with these remarks.

We acknowledge that the cumulative $CO_2$ fluxes as well as the accumulation of dissolved C during waterlogging were higher in the monoliths with the cover crop than without. This was expected but not the main interest of our study. Instead, we were interested in the interaction of water treatment and the cover crop. Against our expectations, the cover crop did not alter soils´ response to waterlogging by providing a substrate supply which in previous research has been shown to promote Fe reduction (e.g. Winkler et al., 2019). We think that this is more important outcome than the increased cumulative $CO_2$ emissions discussed earlier in the Discussion.

**References**

Bergström, L. (1990). Use of lysimeters to estimate leaching of pesticides in

agricultural soils. *Environmental Pollution*, *67*(4), 325–347.

https://doi.org/10.1016/0269-7491(90)90070-S

Chow, A. T.-S., Ulus, Y., Huang, G., Kline, M. A., & Cheah, W.-Y. (2022). Challenges

in quantifying and characterizing dissolved organic carbon: Sampling,

isolation, storage, and analysis. *Journal of Environmental Quality*, *51*(5),

837–871. https://doi.org/10.1002/jeq2.20392

Fernandez-Ugalde, O., Scarpa, S., Orgiazzi, A., Panagos, P., Van Liedekerke, M.,

Marechal, A., & Jones, A. (2022). *LUCAS 2018 soil module: Presentation of*

*dataset and results.* (No. JRC129926). Publications Office of the European

Union. https://data.europa.eu/doi/10.2760/215013

Ginn, B., Meile, C., Wilmoth, J., Tang, Y., & Thompson, A. (2017). Rapid Iron

Reduction Rates Are Stimulated by High-Amplitude Redox Fluctuations in a

Tropical Forest Soil. *Environmental Science & Technology*, *51*(6), 3250–3259.

https://doi.org/10.1021/acs.est.6b05709

Heikkinen, J., Ketoja, E., Nuutinen, V., & Regina, K. (2013). Declining trend of

carbon in Finnish cropland soils in 1974-2009. *Global Change Biology*, *19*(5),

1456–1469. https://doi.org/10.1111/gcb.12137

Herbrich, M., Gerke, H. H., Bens, O., & Sommer, M. (2017). Water balance and

leaching of dissolved organic and inorganic carbon of eroded Luvisols using

high precision weighing lysimeters. *Soil and Tillage Research*, *165*, 144–160.

https://doi.org/10.1016/j.still.2016.08.003

Huang, W., Wang, K., Ye, C., Hockaday, W. C., Wang, G., & Hall, S. J. (2021). High

carbon losses from oxygen-limited soils challenge biogeochemical theory and

model assumptions. *Global Change Biology*, *27*(23), 6166–6180.

https://doi.org/10.1111/gcb.15867

Kronberg, R., Kanerva, S., Koskinen, M., Polvinen, T., Heinonsalo, J., & Pihlatie, M.

(2024). Controlled soil monolith experiment for studying the effects of

waterlogging on redox processes. *Geoderma*, *452*, 117110.

https://doi.org/10.1016/j.geoderma.2024.117110

Lemola, R., Uusitalo, R., Hyväluoma, J., Sarvi, M., & Turtola, E. (2018). *Suomen*

*peltojen maalajit, multavuus ja fosforipitoisuus: Vuodet 1996–2000 ja*

*2005–2009.* Luonnonvarakeskus.

https://jukuri.luke.fi/handle/10024/541851

Lewis, J., & Sjöstrom, J. (2010). Optimizing the experimental design of soil columns in saturated and unsaturated transport experiments. *Journal of Contaminant Hydrology*, *115*(1), 1–13. https://doi.org/10.1016/j.jconhyd.2010.04.001

Niskanen, R. (1989). Extractable aluminium, iron and manganese in mineral soils: I Dependence of extractability on the pH of oxalate, pyrophosphate and EDTA extractants. *Agricultural and Food Science*, *61*(2), Article 2. https://doi.org/10.23986/afsci.72355

Osborne, J. W., & Waters, E. (2002). Four assumptions of multiple regression that researchers should always test. *Practical Assessment, Research, and Evaluation*, *8*(1). https://doi.org/10.7275/r222-hv23

Peltovuori, T., Uusitalo, R., & Kauppila, T. (2002). Phosphorus reserves and apparent phosphorus saturation in four weakly developed cultivated pedons. *Geoderma*, *110*(1), 35–47. https://doi.org/10.1016/S0016-7061(02)00180-5

Räike, A., Kortelainen, P., Mattsson, T., & Thomas, D. N. (2016). Long-term trends (1975–2014) in the concentrations and export of carbon from Finnish rivers to the Baltic Sea: Organic and inorganic components compared. *Aquatic Sciences*, *78*(3), 505–523. https://doi.org/10.1007/s00027-015-0451-2

Rantakari, M., Mattsson, T., Kortelainen, P., Piirainen, S., Finér, L., & Ahtiainen, M. (2010). Organic and inorganic carbon concentrations and fluxes from managed and unmanaged boreal first-order catchments. *Science of The Total Environment*, *408*(7), 1649–1658. https://doi.org/10.1016/j.scitotenv.2009.12.025

Shatz, I. (2024). Assumption-checking rather than (just) testing: The importance of visualization and effect size in statistical diagnostics. *Behavior Research Methods*, *56*(2), 826–845. https://doi.org/10.3758/s13428-023-02072-x

Virtanen, S., Simojoki, A., Knuutila, O., & Yli-Halla, M. (2013). Monolithic lysimeters as tools to investigate processes in acid sulphate soil. *Agricultural Water Management*, *127*, 48–58. https://doi.org/10.1016/j.agwat.2013.05.013

Winkler, P., Kaiser, K., Jahn, R., Mikutta, R., Fiedler, S., Cerli, C., Kölbl, A., Schulz, S., Jankowska, M., Schloter, M., Müller-Niggemann, C., Schwark, L., Woche, S. K., Kümmel, S., Utami, S. R., & Kalbitz, K. (2019). Tracing organic carbon and microbial community structure in mineralogically different soils exposed to redox fluctuations. *Biogeochemistry*, *143*(1), 31–54. https://doi.org/10.1007/s10533-019-00548-7

Winkler, P., Kaiser, K., Thompson, A., Kalbitz, K., Fiedler, S., & Jahn, R. (2018). Contrasting evolution of iron phase composition in soils exposed to redox fluctuations. *Geochimica et Cosmochimica Acta*, *235*, 89–102. https://doi.org/10.1016/j.gca.2018.05.019